# Fair Online Influence Maximization

Xiangqi Wang[†1], Shaokun Zhang[2], Jose Efraim Aguilar Escamilla[3], Qingyun Wu[2], Xiangliang Zhang[1], Jian Kang[4], Huazheng Wang[†3]

[1] *University of Notre Dame*
[2] *Pennsylvania State University*
[3] *Oregon State University*
[4] *University of Rochester*

**Reviewed on OpenReview:** *https://openreview.net/forum?id=T1NjRBI5xs*
[†]Corresponding author: `xwang76@nd.edu`, `huazheng.wang@oregonstate.edu`

## Abstract

Fair influence maximization in networks has been actively studied to ensure equity in fields like viral marketing and public health. Existing studies often assume an offline setting, meaning that the learner identifies a set of seed nodes with known per-edge activation probabilities. In this paper, we study the problem of *fair online influence maximization*, i.e., without knowing the ground-truth activation probabilities. The learner in this problem aims to maximally propagate the information among demographic groups, while interactively selecting seed nodes and observing the activation feedback on the fly. We propose Fair Online Influence Maximization (FOIM) framework that can solve the online influence maximization problem under a wide range of fairness notions. Given a fairness notion, FOIM solves the problem with a combinatorial multi-armed bandit algorithm for balancing exploration-exploitation and an offline fair influence maximization oracle for seed nodes selection. FOIM enjoys sublinear regret when the fairness notion satisfies two mild conditions, i.e., monotonicity and bounded smoothness. Our analyses show that common fairness notions, including maximin fairness, diversity fairness, and welfare function, all satisfy the condition, and we prove the corresponding regret upper bounds under these notions. Extensive empirical evaluations on three real-world networks demonstrate the efficacy of our proposed framework.*

## 1 Introduction

Influence maximization (IM) (Kempe et al., 2003) has been extensively studied with applications in viral marketing (Chen et al., 2010) and public health (Valente & Pumpuang, 2007; Yadav et al., 2018). Classic influence maximization aims to select a set of seed nodes in a social network to maximize the spread of influence (i.e., expected number of influenced nodes). The problem is typically formulated as a stochastic diffusion process (Kempe et al., 2003; Csókás & Vinkó, 2022) in a directed graph, with the edge weights representing the probabilities that the source nodes can activate target nodes.

Recently, there has been an increase in concerns regarding algorithmic fairness motivated by recent studies on fair influence maximization (Farnadi et al., 2020; Tsang et al., 2019; Fish et al., 2019; Becker et al., 2020; Rahmattalabi et al., 2021; Feng et al., 2023). These studies have primarily consider egalitarian utility (the number of influenced nodes across demographic groups) over the classic utilitarian utility (the total number of influenced nodes). With advent of official regulations such as GDPR (European Parliament & Council of the European Union), fair algorithms that control discrimination have received renewed interest. For example, consider a health organization aiming to provide information related to preventive care against a pandemic to as many people as possible. Due to limited resources, the health organization is expected to select a set of influencers (*seed nodes*) who would propagate the vital information to others in a social network. By following egalitarian utility, an ideal choice of influencers would ensure a certain portion of at-risk individuals from some minority (or protected) group will see the information.

---

*The code can be found at `https://github.com/XiangqiWang77/FOIM`

To date, solutions to fair influence maximization almost exclusively assume an offline setting, meaning that the activation probabilities are known a priori (Farnadi et al., 2020; Rahmattalabi et al., 2021; Tsang et al., 2019; Rahmattalabi et al., 2019). However, the activation probabilities may not be fully observable in many real-world applications. Prior works in fair offline influence maximization often obtain this information via simulations over data collected offline without fairness considerations (Rahmattalabi et al., 2021). Consequently, the estimated probabilities could likely be inconsistent for individuals in a minority group due to bias introduced in the offline data selection (e.g., fewer logged activations for individuals from the protected group). Based on the estimated information, this discrepancy can result in sub-optimal seed selection for fair offline influence maximization. This motivates the study of *fair online influence maximization*, whose goal is to estimate the activation probabilities by repeatedly interacting with the network to maximize the information diffusion among demographic groups.

There are three key challenges in fair online influence maximization that must be addressed to develop efficient algorithms. The first challenge deals with the exploration-exploitation tradeoff under a fairness constraint (i.e., across different demographic groups). Second, the combinatorial nature of the influence maximization problem makes finding the optimal seed set an NP-hard problem (Kempe et al., 2003). Thus, it is non-trivial to solve the NP-hard combinatorial problem with a good exploration-exploitation trade-off under the fairness constraint.

The third key challenge to fair online influence maximization is a theoretical challenge. While bandit-based solutions are natural for online influence maximization without fairness considerations, extending their theoretical analysis to fair online scenarios introduces complications. Rigorous regret analyses for bandit-based solutions typically require the reward function to satisfy specific conditions such as submodularity (Hazan & Kale, 2012; Chen et al., 2018; Perrault et al., 2020), monotonicity, and bounded smoothness (Chen et al., 2014; Wang & Chen, 2017a; Wen et al., 2016). However, when considering an egalitarian reward function it is important to ensure the aforementioned characteristics hold for target fair metric.

In this paper, we tackle both algorithmic and theoretical challenges and propose a framework named FOIM for Fair Online Influence Maximization. To tackle the algorithmic challenge, FOIM first runs a combinatorial multi-armed bandit (CMAB) algorithm for parameter estimation and balancing exploration-exploitation on the fly, and then applies an offline fair influence maximization oracle (Tsang et al., 2019; Farnadi et al., 2020) to tackle the NP-hard problem of selecting seed nodes. FOIM allows a wide range of fairness constraints (Tsang et al., 2019; Rahmattalabi et al., 2021), and can incorporate any desired CMAB algorithm. We provide theoretical guarantees in terms of sublinear regret bounds on the FOIM framework under various group fairness notions. In a nutshell, our theoretical results generalize existing analyses of CMAB-based online influence maximization without fairness considerations. A key insight of our analysis shows that the common group-fairness reward functions such as maxmin fairness (Tsang et al., 2019), diversity fairness(Tsang et al., 2019; Rahmattalabi et al., 2021), and welfare functions (Rahmattalabi et al., 2021), all satisfy the monotonicity and bounded smoothness conditions, which allows us to leverage existing analysis of CMAB algorithms and online influence maximization (Chen et al., 2014; Wen et al., 2016; Wu et al., 2019).

- We study the fair online influence maximization problem, whose goal is to improve the quality of activation probabilities and seed node selection on the fly. To the best of our knowledge, this is the first study on *fair* online influence maximization problem.
- We propose a generic FOIM framework that uses a CMAB algorithm for balancing exploration-exploitation and an offline fair influence maximization oracle for seed nodes selection.
- Theoretically, we show that common group fairness notions for influence maximization all satisfy monotonicity and bounded smoothness conditions and prove the corresponding sublinear regret upper bound for each fairness notion.
- We conduct extensive experiments on three real-world datasets. We compare the performance of FOIM with different CMAB algorithms and fairness oracles. Experimental results confirms the effectiveness of FOIM under different fairness notions.

## 2 Related Work

**Fair influence maximization** aims to maximize the influence spread with consideration of fairness (Farnadi et al., 2020; Tsang et al., 2019; Fish et al., 2019; Becker et al., 2020; Rahmattalabi et al., 2021). Existing works on fair influence maximization mainly consider two types of fairness notions: group fairness (Tsang et al., 2019; Rahmattalabi et al., 2021; Farnadi et al., 2020) and individual fairness (Fish et al., 2019; Becker et al., 2020). For group fairness, Tsang et al. (2019) propose a multi-objective Frank-Wolfe based algorithm to ensure maximin fairness and diversity fairness. Rahmattalabi et al. introduce the welfare function which generalizes maximin fairness and diversity fairness, as a fairness notion.

Farnadi et al. (2020) utilize mixed integer programming for fair influence maximization with respect to four different fairness notions (equality, equity, maximin fairness, and diversity fairness). For individual fairness, (Fish et al., 2019) use the welfare function to maximize the minimum probability of individuals receiving the information. Becker et al. achieve individual fairness in influence maximization through randomization. Our work is substantially different from these two lines of research. Compared with Tsang et al. (2019); Rahmattalabi et al. (2021); Farnadi et al. (2020) that study influence maximization in offline setting, we consider an online setting where the activation probabilities are not given, and we aim to estimate the activation probabilities by interacting with the network. Additionally, Fish et al. (2019) and Becker et al. (2020) study individual fairness which is a different fairness notion to the focus of this paper, i.e., group fairness.

**Online influence maximization** assume that the activation probabilities are not known a priori. Several works have been proposed to solve online influence maximization problems with combinatorial multi-armed bandit, including CUCB (Chen et al., 2014), IMLinUCB (Wen et al., 2016) and IMFB (Wu et al., 2019). The key idea is to regard the propagation probability of edges in the graph as the arms of bandits and learn the combinatorial set of arms as super arm in the *online setting*. More specifically, CUCB (Chen et al., 2014), one of the most fundamental CMAB-based online influence maximization algorithms, is an extension of the upper confidence bound (UCB) algorithm (Auer et al., 2002) to the combinatorial setting that achieves asymptotic optimal regret. IMLinUCB (Wen et al., 2016) applies LinUCB (Li et al., 2010), a classic contextual bandit algorithm, for online influence maximization. IMFB (Wu et al., 2019) is a state-of-the-art online influence maximization algorithm, which takes network assortativity and factorizes the edge weights into latent factors. It is worth noting that both IMLinUCB and IMFB are based on CUCB and can achieve asymptotic optimal regret. However, none of them (Chen et al., 2014; Wu et al., 2019; Wen et al., 2016) take fairness into consideration, whereas our work aims to solve online influence maximization under fairness constraints.

## 3 Background

**Influence maximization.** The classical Influence Maximization (IM) problem formulation considers a directed graph $G = (V; E; \mu)$, where $V$ and $E$ represent the set of nodes and edges in the graph, respectively. The graph is further equipped with *edge weights* $\mu = \{\mu_1, \mu_2, \ldots, \mu_{|E|}\}$, $\forall i \in \{1, \ldots, |E|\}$, $\mu_i \in [0, 1]$ for any edge $e_i \in E$. These weights represent the probability that an active node will influence some neighbor node according to some diffusion model. In our case, we exclusively consider the *independent cascade* (IC) model (Kempe et al., 2003).

Formally, solving an IM problem involves choosing some *seed set* $S \in 2^V$ with cardinality constraint $|S| \leq K$ influencing the largest number of nodes under some diffusion model.

$$S^* := \arg\max_{S \in 2^V} I_G(S) \quad \text{s.t. } |S| \leq K, \tag{1}$$

where $S^*$ represents the optimal seed set maximizing influence spread for graph $G$, and $I_G(S)$ is the expected number of influence nodes achieved by seed set $S$.

Influence maximization is a well-known NP-hard problem (Kempe et al., 2003; Nemhauser et al., 1978). This result, mainly due to the combinatorial nature of finding a seed set, makes finding an optimal solution *computationally challenging*. To circumvent this problem, most proposed learning algorithms follow a "two-step" approach. First, the learning algorithm estimates the edge weights $\mu$ using previously collected

data. Then, an algorithm (known as an oracle) chooses an approximate seed set $\hat{S}$. The key to attaining a computationally tractable algorithm is to assume that the oracle algorithm is an $(\alpha, \beta)$-*approximation oracle.*

**Definition 1** $((\alpha, \beta)$-Approximation Oracle (Chen et al., 2014)). An $(\alpha, \beta)$-Approximation Oracle, in the context of Influence Maximization, is defined as any algorithm that can produce a seed set $\hat{S} \in 2^V$ given estimated edge weights $\hat{\mu}$ obeying the following probability:

$$Pr\left[r_\mu(\hat{S}) \geq \alpha \cdot r_\mu(S^*)\right] \geq \beta, \tag{2}$$

where $r_\mu(S)$ is the reward function corresponding to the chosen seed set given edge weights $\mu$.

Intuitively, an $(\alpha, \beta)$-approximation oracle is allowed to produce seed sets that deviate by a fraction $\alpha$ from the theoretical best with probability at least $\beta$. This helps in reducing the computational burden on the oracle algorithm by accepting approximate solutions. For example, in greedy oracle $\alpha$ is set as $1 - \frac{1}{e}$ with $\beta$ set as 1.

**Online IM.** Online IM is referred as not assuming the known per-edge activation probability beforehand, in contrast to offline influence maximization. Specifically, it aims to simultaneously seek the influential nodes (exploitation) and estimate the social network (exploration). And the combinatorial multi-armed bandit (CMAB) framework is a natural choice that can balance the exploration-exploitation trade-off (Wen et al., 2016; Wu et al., 2019; Chen et al., 2014; Vaswani & Lakshmanan, 2015).

Given the graph $G$ and cardinality $K$, an online solution typically follow the following procedure. At each step $t$, the learner first chooses a set of seed nodes $S_t \in V$ with cardinality limit $K$ based on historical information, by performing offline IM (regardless of fair or not) algorithms according to the current estimation of the graph. Here, these algorithms regard the solution obtained as an ORACLE. We denote it as $S_t = \text{ORACLE}(G, \hat{\mu}_t, K)$, where $\hat{\mu}_t$ represents the estimation of per-edge activation probability at step $t$. Then, the learner updates the estimation for the graph with $S_t$ as the seed set. $r_\mu(S_t)$ is utilized to represent the corresponding reward, which might be modified with fair constraints. The learner's objective is to minimize the expected cumulative regret by finding a sequence of seed nodes $\hat{S}$ over a finite $T$ rounds of interactions:

$$Regret = \sum_{t=1,\ldots,T} \left[r_\mu\left(S_t^*\right) - r_\mu\left(\hat{S}_t\right)\right]. \tag{3}$$

Note that the reward function $r_\mu(S_t)$ is a general notation that can represent not only influence spread but also other reward functions such as fair IM reward.

**Fairness in IM.** Given its applicability to real-world problems, the IM problem highlights *fairness* as a major concern in algorithm design. Previous literature has introduced various fairness metrics like maximizing minority group welfare in graph through IM process, but incorporating fairness in an online setting remains challenging due to the lack of consensus on effective fairness integration. This paper presents an algorithmic design framework that generalizes fairness measures while enabling theoretical regret analysis within the Combinatorial Multi-armed Bandit (CMAB) framework, specifically designed for online applications.

## 4   The Fair Online Influence Maximization (FOIM) Framework

We begin the presentation of our novel FOIM framework by extending our formulation of influence maximization (IM) to include fairness. Starting from a set of seed nodes $S$, we use $I_{G,C_i}(S)$ to denote the expected number of influenced nodes that belong to not necessarily disjoint group (or community) $C_i$. In one diffusion process[†], we use the set $u = \{u_1, u_2, \ldots, u_{|V|}\}$ to represent the probability of each node being activated. We employ $u^{C_i} = \frac{I_{G,C_i}(S)}{|C_i|}$ to represent the expected fraction of the nodes being activated in group $C_i$. Additionally, given graph $G$ and any feasible set of seed nodes $S$, we use $E_S$ to denote the set of edges in the subgraph induced from graph $G$, where the subgraph is formed by seed nodes $S$. We also employ $\tilde{E}_S$ to represent the set of edges reachable from seed set $S$ in one IC diffusion process.

---

[†]We refer to a diffusion process as a single cycle of influence within network $G$, where influenced nodes are allowed to attempt to influence neighbor nodes once.

Formally, given a graph $G$, *fair influence maximization* aims to find a set of nodes $S^*$ under a cardinality constraint $K$, which maximizes the influence spread, where $I_{G,C_i}(S)$ denotes the influenced nodes by some seed set $S$ with respect to community $C_i$.

$$S^* \coloneqq \arg \max_{|S| \leq K} \sum_{C_i \in C} I_{G,C_i}(S) \tag{4}$$

Using a slight abuse of notation, we let $I_G(k)$ be the maximum number of influence nodes that can be achieved with cardinality constraint $k$

$$I_G(k) = \sum_{C_i \in C} I_{G,C_i}(S^*) \ \ \text{s.t.} \ \ |S^*| \leq k. \tag{5}$$

### 4.1 Fairness Constraints

In this section, we demonstrate how our FOIM framework effectively incorporates various fairness constraints, covering most of the fairness aspects introduced in prior literature. After presenting the studied fairness metrics, we propose the FOIM algorithm. In addition, we provide discussion of FOIM on TPM condition in Appendix A.13

**FOIM with maximin fairness.** Maxmin fairness proposed by Tsang et al. (2019); Diana et al. (2021) measures the influence of the minimal group in the influence maximization algorithm. Given any feasible seed nodes $S$, the IM reward function with maximin fairness constraint is

$$r_\mu^{maxmin}(S) = \min_{\forall C_i \in C} \frac{I_{G,C_i}(S)}{|C_i|} \tag{6}$$

where $\frac{I_{G,C_i}(S)}{|C_i|}$ is the proportion of the expected number of influenced nodes for group $C_i$ to its population, and $\mu$ denotes the edge weights in the graph. Searching the seed nodes with this reward function could be described as

$$\hat{S} = \arg \max_{S \in S} \{r_\mu^{maxmin}(S)\}. \tag{7}$$

**FOIM with diversity fairness.** The key idea of the diversity fairness constraint from Tsang et al. (2019) is that no group benefits more by leaving the IM game and allocating its proportional resources internally. For the $i$-th group $C_i$, we define $k_i = \lceil K|C_i|/|C| \rceil$ as the fair allocation of influenced nodes in $C_i$ based on group size, where $K$ is the seed budget. Let $G[C_i]$ denote the subgraph induced by nodes in $C_i$, while keeping their original connections in $G$. To satisfy the diversity fairness constraint, the selected seed nodes $S$ should satisfy

$$I_{G,C_i}(S) \geq I_{G[C_i]}(k_i), \ \ \forall C_i \in C. \tag{8}$$

Then given any feasible seed nodes $S$ and graph $G = (V; E; \mu)$, the IM reward function with diversity fairness is defined as

$$r_\mu^{diversity}(S) = \begin{cases} \sum_{C_i \in C} I_{G,C_i}(S) & \text{if Equation (8) holds} \\ 0 & \text{otherwise.} \end{cases} \tag{9}$$

And searching the seed nodes for Eq. (9) is

$$\hat{S} = \arg \max_{S \in S} (r_\mu^{diversity}(S)). \tag{10}$$

**FOIM with welfare function.** The welfare function (Rahmattalabi et al., 2021) is a reward function that generalizes multiple fairness notions in IM, including *maximin fairness* and *diversity fairness*. It encourages uniform influence among different groups, with the trade-off between fairness and efficiency adjusted by a hyperparameter $\alpha$. Mathematically, the welfare reward function is

$$r_\mu^{welfare}(S) = \begin{cases} \sum_{c \in C} \left[ |c| \cdot (u^c)^\alpha \right] / \alpha, & \text{if } \alpha \neq 0 \\ \sum_{c \in C} |c| \cdot \log(u^c), & \text{if } \alpha = 0 \end{cases} \tag{11}$$

where $u^c$ denotes the expected fraction of the influenced vertices of community $c$, and $u^c \geq 0$, $\forall c \in C$. Notably, when $\alpha \to -\infty$, equation (11) is equivalent to the reward function for maximin fairness (Equation (6)).

## 4.2 FOIM Algorithm

Now, we introduce the proposed Fair Online Influence Maximization (FOIM) framework, which solves fair online influence maximization problem with a wide range of fair rewards. Notably, the proposed FOIM framework is a general framework that supports the integration of *any* combinatorial multi-armed bandit (CMAB) algorithm under certain conditions. In Section 5, we prove that these conditions are naturally satisfied in common fair rewards and demonstrate that FOIM could achieve sublinear regret (Chen et al., 2014; Kveton et al., 2014). We present the FOIM framework in Algorithm 1.

---

**Algorithm 1:** Fair Online Influence Maximization (FOIM)

---
1: **Input** Graph $G$, time budget $T$, seed nodes size $K$, reward function $r_\mu(S)$, edge weight estimator of CMAB $\mathcal{A}(G, \hat{\mu}_t, r_t)$, fair oracle FairORACLE$(G, \hat{\mu}_t, K)$.
2: **Initialize** $t \leftarrow 0$
3: **for** t in $\{1, \ldots, T\}$ **do**
4:     $\hat{\mu}_t \leftarrow \mathcal{A}(G, \hat{\mu}_{t-1}, r_{t-1})$ `// estimate edge weights`
5:     $S_t$=FairORACLE$(G, \hat{\mu}_t, K)$ `// obtain seed nodes`
6:     $r_t \leftarrow r_\mu(S_t)$ `// obtain rewards`
7: **end for**

---

For instance, if CUCB (Chen et al., 2014) serves as backbone of $\mathcal{A}$, $\hat{\mu}$ is estimated independently with UCB (Stone, 2010) algorithm at each time step. If $\mathcal{A}$ belongs to IMLinUCB (Wen et al., 2016), $\hat{\mu}$ is estimated with the contextual bandit LinUCB algorithm with an additional input of edge features. Another worth mentioned input is the fair oracle FairORACLE$(G, \hat{\mu}_t, K)$, which identifies the optimal seed nodes based on the current estimated edge weights. Actually, any fair offline CMAB algorithms with $(\alpha, \beta)$ approximation guarantee could be utilized, such as multi-objective Frank-Wolfe algorithm (Tsang et al., 2019) and Mixed Integer Programming (MIP) (Farnadi et al., 2020).

Given the graph $G$ and cardinality $K$, FOIM runs with the following procedure: at each step $t$, CMAB algorithm $\mathcal{A}$ estimates the edge weights $\hat{\mu}_t$ according to the regret $r_{t-1}$ and $\hat{\mu}_{t-1}$ from at last time step. Then, FairORACLE obtains the seed nodes based on the current estimated edge weights $\hat{\mu}_t$ and calculate the regret for the obtained seed nodes. The procedure repeats until the given time budget $T$ is reached. In addition, we give the time complexity analysis of FOIM framework in Appendix A.12.

## 5 Theoretical Analysis

It has been demonstrated that a CMAB algorithm will achieve a sublinear regret bound when the corresponding reward function satisfies both monotonicity and bounded smoothness conditions (Chen et al., 2014; Wu et al., 2019; Wen et al., 2016). Here we first show the monotonicity and bounded smoothness conditions for fairness metrics below.

**Condition 1.** *Given graph $G = (V; E; \mu)$, we require the following two conditions on the fair online IM reward $r_\mu(S)$ for FOIM:*

- ***Monotonicity.*** *Given any feasible seed nodes $S$, the expected reward $r_\mu(S)$ is monotonically nondecreasing with respect to the edge weights of the graph, i.e., if $\forall i \in \{1, \ldots, |E|\}$, $\mu_i \leq \mu_i'$, then we have $r_\mu(S) \leq r_{\mu'}(S)$.*
- ***Bounded smoothness.*** *There exists a continuous, strictly increasing (thus invertible) function $f(\cdot)$ with $f(0) = 0$, called bounded smoothness function, such that for two $\mu$ and $\mu'$, and for any $\Lambda > 0$, we have $|r_\mu(S) - r_{\mu'}(S)| \leq f(\Lambda)$, if $\max\limits_{i \in \bar{E}_S} |\mu_i - \mu_i'| \leq \Lambda$.*

Note that FOIM is a general framework for fair online influence maximization based on CMAB algorithms. Specifically, we prove that the reward function with multiple fairness constraints satisfies the *monotonicity* and *bounded smoothness* conditions, implying the generality of FOIM being the solution to fair online influence maximization with different fairness constraints.

### 5.1 Fairness Analaysis

**Maximin fairness.** In Lemma 1, we show that the reward functions for fair online influence maximization with maximin fairness satisfy Condition 1, i.e., monotonicity and bounded smoothness.

**Lemma 1.** *Given graph $G = (V; E; \mu)$, the IM reward function with maximin fairness constraint satisfies monotonicity. It also satisfies bounded smoothness with smoothness function $f(x) = |E|x$.*

*Proof.* We first prove the monotonicity and then show the reward function satisfies bounded smoothness.

**P1 − Monotonicity.** Given two graph $G$ and $G'$ with the set of edge weights $\mu$ and $\mu'$ respectively, if $\forall i \in \{1, \ldots, |E|\}, \mu_i \leq \mu_i'$, it is easy to have $\forall C_i \in C$:

$$I_{G,C_i}(S) \leq I_{G',C_i}(S). \tag{12}$$

Let $C_i$ and $C_{i'}$ be the groups receiving the minimum proportional influence in $G$ and $G'$ respectively:

$$i = \arg\min_{C_j \in C} \frac{I_{G,C_j}(S)}{|C_j|}, \quad i' = \arg\min_{C_j \in C} \frac{I_{G',C_j}(S)}{|C_j|}. \tag{13}$$

Regardless of whether $i = i'$ or $i \neq i'$, we have:

$$\frac{I_{G,C_i}(S)}{|C_i|} \leq \frac{I_{G,C_{i'}}(S)}{|C_{i'}|} \leq \frac{I_{G',C_{i'}}(S)}{|C_{i'}|}. \tag{14}$$

Therefore:

$$r_\mu^{maxmin}(S) = \frac{I_{G,C_i}(S)}{|C_i|} \leq \frac{I_{G',C_{i'}}(S)}{|C_{i'}|} = r_{\mu'}^{maxmin}(S). \tag{15}$$

Thus, the monotonicity property $r_\mu^{maxmin}(S) \leq r_{\mu'}^{maxmin}(S)$ is proven.

**P2 − Bounded smoothness under $f(x) = |E|x$.** Without loss of generality, suppose there exists graph $G$ and $G'$ with edge weights $\mu$ and $\mu'$, respectively, $\forall i \in E \ \mu_i' = \mu_i + \Lambda, \Lambda > 0$. According to monotonicity of the reward function, we have $r_\mu^{maxmin}(S) \leq r_{\mu'}^{maxmin}(S)$ for any feasible seed nodes $S$. For any edge $i = (v, u) \in \tilde{E}_S$, the increase of activation probability between $v$ and every node reachable from $v$ in $G'$ brought by $i$-th edge is **at most** $\Lambda$ compared with it in graph $G$.

Let $C_{\min}$ and $C_{\min}'$ be the demographic groups receiving the minimum proportional influence in $G$ and $G'$, respectively. If $C_{min} = C_{min}'$, the total increase of influence nodes in $C_{min}$ bringing by any edge in $\tilde{E}_S$ is at most $|C_{min}|\Lambda$. So the total increase of IM reward function with maximin fairness constraint in $G'$ is $\frac{|C_{min}|\Lambda}{|C_{min}|} = \Lambda$. If $C_{min} \neq C_{min}'$, the total increase of IM reward function must be smaller than $\frac{I_{G',C_{min}}(S)}{|C_{min}|} - \frac{I_{G,C_{min}}(S)}{|C_{min}|}$ considering $C_{min} \neq C_{min}'$, and group $C_{min}$ achieves the smallest number of influenced nodes as portion to its population in $G$. Since $\frac{I_{G',C_{min}}(S)}{|C_{min}|} - \frac{I_{G,C_{min}}(S)}{|C_{min}|}$ is bounded by $\Lambda$, the reward increase is also bounded by $\Lambda$.

Here we let $s = |\tilde{E}_S|$. The total reward increase bringing by edges set $\tilde{E}_S$ is at most $s\Lambda \leq |E|\Lambda$. Since increasing the weights in edges that either cannot be reached from seed nodes $S$ or belong to $E_S$ does not lead to an increase in reward, the reward improvement solely benefits from $\tilde{E}_S$. We conclude that the IM utility function with maximin fairness satisfies bounded smoothness with $f(x) = |E|x$. □

**Diversity fairness.** We show in Lemma 2 that the reward function for fair online influence maximization with diversity fairness satisfies Condition 1, i.e., monotonicity and bounded smoothness.

**Lemma 2.** *Given graph $G = (V; E; \mu)$, the reward function for fair online influence maximization with diversity fairness constraint satisfies monotonicity. It also satisfies bounded smoothness with smoothness function $f(x) = |E||V|x + |V|$.*

*Proof.* We first prove the monotonicity and then show the reward function satisfies bounded smoothness. In addition, we need to use Lemma 3 to assist our proof and Lemma 3 has detailed proof referred in Appendix A.9.

**Lemma 3.** *If the diversity constraint is satisfied with $\mu$, the constraint will not be violated after the lifting of $\mu$ (adding $\Lambda$ to $\mu$)*

**P1 − Monotonicity.** Given two graph $G$ and $G'$ with the set of edge weights $\mu$ and $\mu'$ respectively, if $\mu_i \leq \mu'_i, \ \forall j \in \{1, \ldots, |E|\}$ holds, it is trivial that for class $C_i$

$$I_{G,C_i}(S) \leq I_{G',C_i}(S) \qquad I_{G[C_i]}(k_i) \leq I_{G'[C_i]}(k_i). \tag{16}$$

(1) Suppose both $G$ and $G'$ satisfy diversity constraint given $S$. By Equation (16) and Equation (9), we have

$$\sum_{C_i \in C} I_{G,C_i}(S) = r_\mu^{diversity}(S) \leq r_{\mu'}^{diversity}(S) = \sum_{C_i \in C} I_{G',C_i}(S) \tag{17}$$

(2) Suppose at least one of $G$ and $G'$ violates the diversity constraint given $S$. By Equation (16), it is impossible that $G$ satisfies the diversity constraint while $G'$ violates (see more details in Lemma 3). Then

- If both $G$ and $G'$ violate the diversity constraint, we have that $r_\mu^{diversity}(S) = r_{\mu'}^{diversity}(S) = 0$;
- If $G$ violates the diversity constraint while $G'$ satisfies, we have that $r_\mu^{diversity}(S) = 0$ and $r_{\mu'}^{diversity}(S) > 0$.

Thus, the reward function is monotone, i.e., $r_\mu^{diversity}(S) \leq r_{\mu'}^{diversity}(S)$.

**P2 − Bounded smoothness under $f(x) = |E||V|x + |V|$.** The proof is irregular as it results in a bounded smoothness function of the form $f(x) = kx + B$. Therefore, its detailed proof and the rationale behind this form are provided in Appendix A.10.

$\square$

**Welfare function.** With that in mind, we show in Lemma 4 that the reward function for fair online influence maximization with welfare function as fairness constraint satisfies monotonicity and bounded smoothness (Condition 1).

**Lemma 4.** *Given graph $G = (V; E; \mu)$, the IM reward function with welfare function as fairness constraint satisfies monotonicity. It also satisfies bounded smoothness with the smoothness function as*

$$f(x) = \begin{cases} [|E||V|((1+x)^\alpha - 1)]/\alpha, & if \ \alpha \geq 1 \\ [|E||V|x^\alpha]/\alpha, & if \ 0 < \alpha \leq 1 \ ^\ddagger \\ [|E||V|((\epsilon + x)^\alpha - \epsilon^\alpha)]/\alpha, & if \ \alpha < 0, \end{cases} \tag{18}$$

*where $\epsilon = \min(\{u^c | \forall c \in C\})$.*

*Proof.* We first prove the monotonicity and then show the reward function satisfies bounded smoothness.

**P1 − Monotonicity.** As shown in Rahmattalabi et al. (2021), the welfare function is monotone with respect to the expected fraction of influenced nodes in each group. For graphs $G$ and $G'$ with edge weights $\mu$ and $\mu'$, if $\mu_i \leq \mu'_i, \forall i \in \{1, \ldots, |E|\}$, then $u^c \leq u'^c, \forall c \in C$, implying $r_\mu^{welfare}(S) \leq r_{\mu'}^{welfare}(S)$. Thus, increasing edge weights preserves monotonicity of the reward function.

---

$^\ddagger f(x) = |E||V|[\log(\epsilon + x) - \log \epsilon]$ if $\alpha = 0$; the analysis is similar to $\alpha < 0$ case.

**P2** – **Bounded smoothness under Equation** (18)**.** Given graph $G$ and any feasible seed nodes $S$, we use $E_S$ to denote the set of edges in the subgraph induced from graph $G$. We employ $\tilde{E}_S$, to represent the set of edges reachable from seed set $S$ in the expected IC diffusion process. Suppose there exists graph $G$ and $G'$ with edge weights $\mu$ and $\mu'$ respectively, where $\mu'_i = \mu_i + \Lambda$, $\forall i \in E$, $\Lambda > 0$. The reward increase in $G'$ solely benefits from the weight improvement in edges set $\tilde{E}_S$. For any edge $i \in \tilde{E}_S$, the total increase of the reward function bought by edge $i$ is **at most** $\sum_{c \in C} (|c|(u^c + \Lambda)^\alpha)/\alpha - \sum_{c \in C} (|c|(u^c)^\alpha)/\alpha$ compared with it in graph $G$. Due to space limitation, we defer the detailed discussion of the upper bound $\sum_{c \in C} (|c|(u^c + \Lambda)^\alpha)/\alpha - \sum_{c \in C} (|c|(u^c)^\alpha)/\alpha$ in Appendix A.11. In short summary, by Appendix A.11, we have the following bounded smoothness function

$$f(x) = \begin{cases} |E||V|((1+x)^\alpha - 1)/\alpha & \text{if } \alpha \geq 1 \\ |E||V|x^\alpha/\alpha & \text{if } 0 < \alpha < 1 \\ |E||V|((\epsilon + x)^\alpha - \epsilon^\alpha)/\alpha & \text{if } \alpha < 0 \end{cases} \tag{19}$$

where $\epsilon = \min(\{u^c | \forall c \in C\})$. [§]

$\square$

**Fairness notion incompatible with FOIM.** While compatible with most fairness metrics in the literature, it should be noted that FOIM may not be universally applied to all fairness notions for influence maximization due to the requirement of theoretical analysis. One example is equity defined in Farnadi et al. (2020). Intuitively, equity asks for the expected number of influenced nodes within a group, denoted as $C_i$, is proportionate to its population ratio

$$\frac{E\left[I_{G,C_i}(S)\right]}{E\left[I_G(S)\right]} \approx \frac{|C_i|}{|V|} \tag{20}$$

This fairness notion violates theoretical requirement of *monotonicity*. Consider $\mu = \mu' + \Lambda$ where the increase of weight is on edges within the subgraph formulated by group $C_i$ and the increased spread does not propagate to groups other than $C_i$. Consequently, only the spread in $C_i$ would increase, while the spread in other groups remains the same. Thus, $E\left[I_{G,C_i}(S)\right]$ would have a larger share in $E\left[I_G(S)\right]$, yet $\frac{|C_i|}{|V|}$ is the same and violates Equation (20).

### 5.2 FOIM Fair Regret Analysis

In this section, we analyze the regret bound for the proposed FOIM framework. We investigate the regret for FOIM with CUCB, which resembles the analysis of other online IM algorithms such as IMLinUCB (Wen et al., 2016) and IMFB (Wu et al., 2019). In particular, we explore how regret will change when the fair IM reward is employed. We primarily present the problem-dependent regret bounds (Chen et al., 2014; Audibert & Bubeck, 2009; Audibert et al., 2011) under different fairness notions. We define $\Delta_{i,\min}$ and $\Delta_{i,\max}$ as the minimum and maximum regret gaps among all bad seed sets that contain edge $i$, respectively, where each regret gap is given by $\Delta_{i,j} = \alpha \cdot r_\mu(S^*) - r_\mu(S^j_{B,i})$.

**Theorem 1** (**Maximin fairness regret**)**.** *According to Lemma 1, the regret of CUCB under maximin fairness is*

$$\sum_{i \in \{1,\ldots,|E|\}, K_i > 0} \frac{24 \cdot |E|^2 \cdot \ln T}{\Delta^i_{\min} \cdot p_i} + \left(\frac{\pi^2}{2} + 1\right) \cdot |E| \cdot \Delta_{\max} \tag{21}$$

Note that the problem-dependent regret is in $\mathcal{O}(\ln T / \Delta)$. The proof is detailed in Appendix A.4.

**Theorem 2** (**Diversity fairness regret**)**.** *According to Lemma 2, the regret of CUCB under diversity fairness is*

$$\sum_{i \in \{1,\ldots,|E|\}, K_i > 0} \left[ \frac{12 \ln T |E|^2 |V|^2}{(\Delta^i_{min} - |V|)^2 p_i} \Delta^i_{min} + \frac{12 \ln T |E|^2 |V|^2}{p_i(|V| - \Delta^i_{max})} \right] + \left(\frac{\pi^2}{2} + 1\right) |E| \Delta_{\max} \tag{22}$$

---

[§]Notably, when $\alpha = 1$, the function $f(x)$ simplifies to $f(x) = |E||V|x$, which corresponds to the bounded smoothness function of the original IM problem, making the proof benign.

This regret is also in $\mathcal{O}(\ln T/\Delta)$. For detailed proof, please refer to Appendix A.3.

**Theorem 3** (**Welfare function regret**). *According to Lemma 4, we analyze the regret of CUCB under welfare function under different $\alpha$ values.*

*When $1 \leq \alpha \neq 2$, we have the regret as*

$$
\left(\frac{\pi^2}{2}+1\right)|E|\Delta_{\max} + \sum_{\substack{i\in\{1,\dots,|E|\}\\ K_i>0}} \max\left\{\frac{24|V|^2|E|^2\ln T}{\Delta_{\min}^i p_i}, \frac{12\ln T\Delta_{min}^i}{\left[\frac{\alpha}{(2^\alpha-1)|E||V|}\Delta_{min}^i\right]^{\frac{2}{\alpha}}p_i} + \frac{12\alpha\ln T\left[(\Delta_{max}^i)^{1-\frac{2}{\alpha}}-(\Delta_{min}^i)^{1-\frac{2}{\alpha}}\right]}{(\alpha-2)p_i\left[\frac{\alpha}{(2^\alpha-1)|E||V|}\right]^{\frac{2}{\alpha}}}\right\}
$$
(23)

*When $\alpha = 2$, we have the regret as*

$$
\sum_{i\in\{1,\dots,|E|\}, K_i>0}\left[\frac{18|E||V|\ln T\cdot}{p_i} + \frac{36|E||V|\ln T(\Delta_{max}^i-\Delta_{min}^i)}{p_i}\right] + \left(\frac{\pi^2}{2}+1\right)|E|\Delta_{\max}
$$
(24)

*When $0 < \alpha < 1$, we have regret bound as*

$$
\sum_{i\in\{1,\dots,|E|\}, K_i>0}\left[\frac{12\ln T\cdot\Delta_{min}^i}{\left[\frac{\alpha}{|E||V|}\Delta_{min}^i\right]^{\frac{2}{\alpha}}p_i} + \frac{12\alpha\ln T\left[(\Delta_{max}^i)^{1-\frac{2}{\alpha}}-(\Delta_{min}^i)^{1-\frac{2}{\alpha}}\right]}{(\alpha-2)p_i\left[\frac{\alpha}{|E||V|}\right]^{\frac{2}{\alpha}}}\right] + \left(\frac{\pi^2}{2}+1\right)|E|\Delta_{\max}
$$

*When $\alpha < 0$, we have regret bound as*

$$
\left(\frac{\pi^2}{2}+1\right)|E|\Delta_{\max} + \sum_{\substack{i\in\{1,\dots,|E|\}\\ K_i>0}} \max\left\{\frac{24\ln T}{p_i}\Delta_{max}, \frac{12\ln T\cdot\Delta_{min}^i}{\left[\frac{\alpha}{[(1+\epsilon)^\alpha-\epsilon^\alpha]|E||V|}\Delta_{min}^i\right]^{\frac{-2}{\alpha}}p_i} + \frac{12\alpha\ln T\left[(\Delta_{min}^i)^{1+\frac{2}{\alpha}}\right]}{(\alpha+2)p_i\left[\frac{\alpha}{[(1+\epsilon)^\alpha-\epsilon^\alpha]|E||V|}\right]^{\frac{-2}{\alpha}}}\right\}
$$
(25)

For the detailed proof, please refer to appendix A.5.

# 6 Empirical Results

We assess FOIM with CMAB backbones under multiple fairness metrics, detailing datasets and bandit variants, then report results that confirm superior performance and show the algorithm accurately recovers edge weights—key to optimal influence maximization.

## 6.1 Cumulative Regret Comparison

**Datasets.** We conduct our experiments on five real-world networks[¶], namely NBA, German, Pokec-z, and bail, which are commonly used for fair graph learning (Dai & Wang, 2020; Ma et al., 2022), as well as YouTube. Detailed dataset statistics and descriptions are deferred in Table 1 and Appendix A.7.

| Dataset | Nodes | Edges | Sensitive Attribute | Activation Probability |
|---------|-------|-------|---------------------|------------------------|
| NBA | 400 | 16,570 | Country | Jaccard index of nodes |
| German | 1,000 | 24,970 | Age | Dice ratio of nodes |
| Pokec-z | 6,185 | 15,321 | Sex | Cosine similarity of nodes |
| Bail | 18,876 | 403,977 | Race | Cosine similarity of nodes |
| YouTube | 1,134,890 | 2,987,624 | Community | Cosine similarity of nodes |

Table 1: Dataset statistics

**CMAB algorithms.** We compare four different combinatorial multi-armed bandit (CMAB) algorithms in FOIM, including CUCB (Chen et al., 2014), $\epsilon$-greedy (Kempe et al., 2003), IMLinUCB (Wen et al., 2016), and IMFB (Wu et al., 2019).

- **CUCB** (Chen et al., 2014). It estimates the per-edge activation probability independently and uses the upper confidence bound of the estimation to balance exploitation and exploration.

---

[¶]https://github.com/yushundong/Graph-Mining-Fairness-Data/tree/main/dataset/

- **$\epsilon$-greedy** (Kempe et al., 2003). It estimates the per-edge activation probability independently and uses $\epsilon$-greedy to balance exploitation (with probability $1 - \epsilon$) and exploration (with probability $\epsilon$).
- **IMLinUCB** (Wen et al., 2016) (LinUCB). It is a model-independent contextual bandit algorithm that requires features on the receiving nodes as input to estimate the reachability between node pairs.
- **IMFB** (Wu et al., 2019). It is a factorization bandit algorithm that factorizes the activation probability on the edges into latent factors on the corresponding nodes.

**Offline oracle.** In addition to the CMAB algorithm, another key component in the propose FOIM framework is the fair oracle, which is the fair offline influence maximization algorithm for seed node selection. In our experiments, we use two different fair oracles, namely multi-objective Frank-Wolfe (FW) algorithm (Tsang et al., 2019) (with suffix -FW) and mixed integer programming (MIP) (Farnadi et al., 2020) (with suffix -MIP). To demonstrate the efficacy of using fair oracle, we use degree discount (Chen et al., 2009) (i.e., methods without suffix), which is an offline influence maximization without fairness consideration, as oracle.

**Reward and regret.** For maximin fairness, we set the reward function as Equation (6); for diversity fairness, we set the reward function as Equation (9); and for welfare function, the reward function is defined as in Equation (11). The cumulative regrets for all methods are calculated using Equation (3). Optimal reward is calculated by the fair IM oracle using ground-truth activation probabilities.

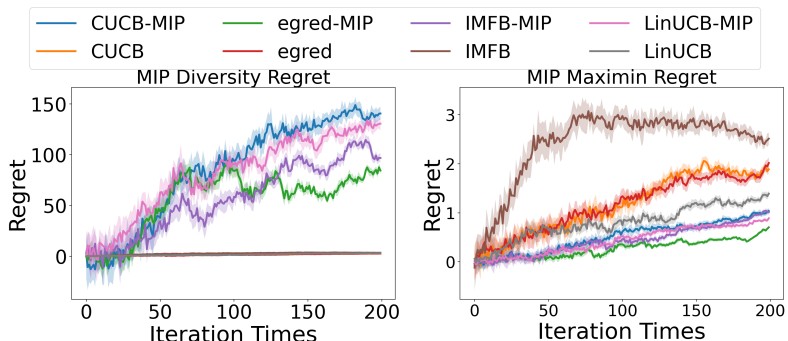

Figure 1: Cumulative regrets under maximin fairness and diversity fairness in NBA. Methods with suffix -MIP use mixed integer programming (Farnadi et al., 2020) as offline oracle with fairness consideration. Methods without suffix use degree discount (Chen et al., 2009) as the offline oracle.

**Hyperparameter settings.** In all experiments, the dimensionalities for the susceptibility factor in IMLinUCB and the latent factors in IMFB are both set to 20. The seed set volume is set to $K = 300$ for Pokec-z and bail dataset, and $K = 10$ for NBA and $K = 50$ for German dataset respectively.

**Activation probability.** For all datasets, we select the first 8 features for each dataset with the sensitive attribute included. Then we initialize the activation probability with the heuristics as presented in Table 1 between node features.

**Regret with fairness constraints under FW oracle.** In Figure 3 and Figure 2, we present the experimental results on the effectiveness of FOIM in reducing the cumulative regret for maximin fairness, diversity fairness, and welfare function ($\alpha = 2$, $\alpha = 0.5$, and $\alpha = -2$) with multi-objective Frank-Wolfe (FW) algorithm (Tsang et al., 2019) oracle. From the figures, we observe that the cumulative regret with fairness constraints achieved under the fair oracle is significantly lower compared to other CMAB frameworks. This highlights the effectiveness of FOIM in improving the fairness of the algorithms.

**Regret with fairness constraints under MIP oracle (Farnadi et al., 2020).** We also test the regret of FOIM with MIP (Farnadi et al., 2020) as oracle on maximin fairness and diversity fairness. Figure 1 presents the regret with MIP oracle on the NBA dataset. From the figure, we obtain similar observation that FOIM with MIP can achieve lower cumulative regret.

## 6.2 Estimation Error Analysis

The empirical findings presented in our study provide substantial evidence supporting the accuracy and integrity of our fair experiment. In order to evaluate the performance of different experiments conducted under fair or submodular oracles, we quantitatively analyze the estimation error. This error metric captures the absolute discrepancy between the estimated activation probability $p_{e,t}$ and ground-truth probability $p_e$ over observed edges. Thus we are estimating the trend of $|\hat{p}_{e,t} - p_e|$.

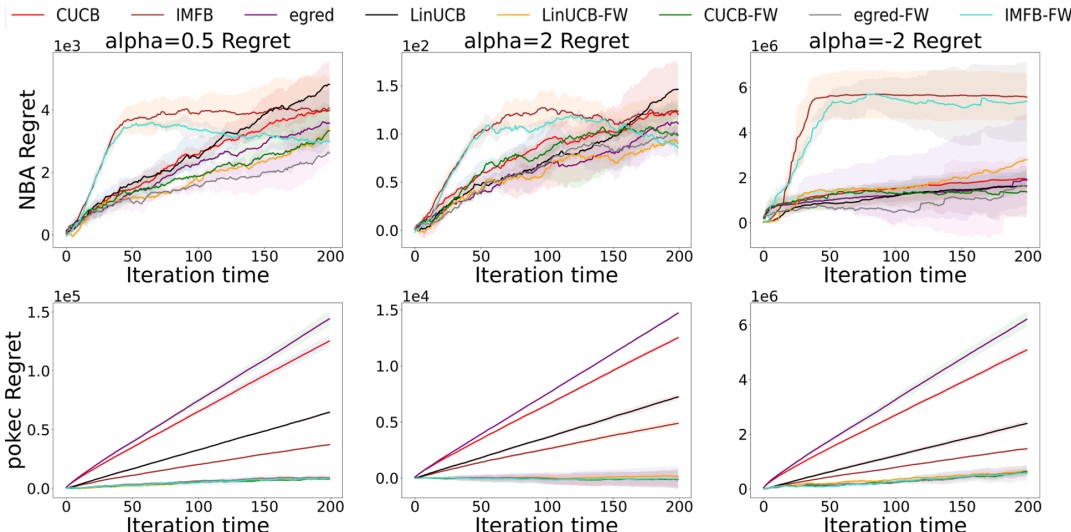

Figure 2: Cumulative regrets and their standard deviation under welfare function ($\alpha = -2, 2, 0.5$) in Pokec-z and NBA. Methods with suffix -FW use the multi-objective Frank-Wolfe algorithm (Tsang et al., 2019) as offline oracle with fairness consideration. Methods without suffix use degree discount (Chen et al., 2009) oracle.

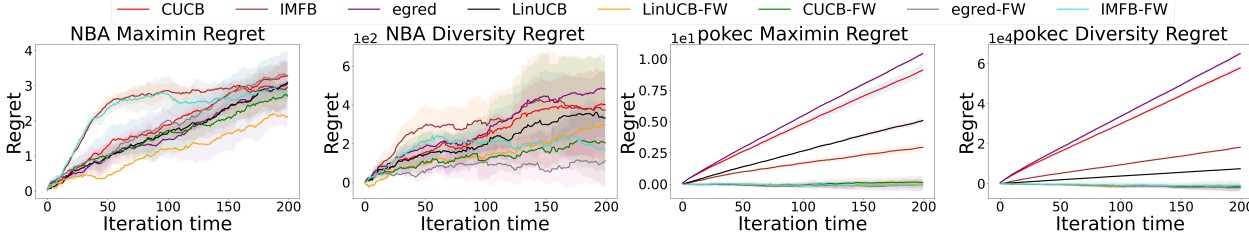

Figure 3: Cumulative regrets and their standard deviation under maximin fairness, diversity fairness in NBA and Pokec-z dataset. Methods with suffix -FW use the multi-objective Frank-Wolfe algorithm (Tsang et al., 2019) as offline oracle with fairness consideration. Methods without suffix use degree discount (Chen et al., 2009) as the offline oracle.

Upon careful examination of the results in Figure 4, it becomes apparent that the disparity in estimation error between the fair oracle and the submodular oracle is not significant. This observation suggests that the fair oracle performs nearly as effectively as submodular oracle like degree discount. Consequently, this comparative analysis lends support to the validity of our fair experiment and lends credence to the notion that the fair modification implemented in our study is both reasonable and efficacious.

### 6.3 Additional Experiments

In Appendix A.8.2, we provide additional results on German, Bail and Youtube datasets with similar observations to prove fair regret smaller result with FOIM and scalability to millions of nodes. In addition, we also provide offline and online influence comparison in Appendix A.8.6 and trade-off between fair metrics and influence in Appendix A.8.7. We also additionally provide theoretical FOIM time complexity analysis and real-time running result in Appendix A.12 as additional empirical proof.

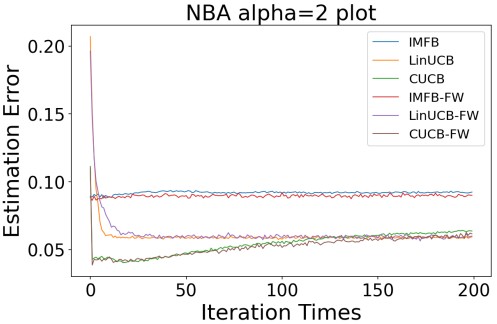

Figure 4: Estimation error of activation probabilities in NBA.

# 7 Conclusion and Future Work

To promote equity and inclusivity in the online environment, we propose the FOIM framework, which combines fair offline influence maximization solutions with online combinatorial bandit models. We demonstrate the feasibility of using fairness constrained reward and oracle in combinatorial bandit algorithms. Additionally, we provide a thorough analysis of regret bounds of FOIM using the CUCB algorithm as an example. Empirical comparisons on three real-world datasets highlight the superior performance in terms of regret under fairness notions, and we validate the efficiency and accuracy through estimation error. However, it is important to note that our analysis currently focuses on group fairness notions. We consider to explore individual fairness as future work.

# Acknowledgements

Jose Efraim Aguilar Escamilla and Huazheng Wang are supported in part by National Science Foundation under grant IIS-2403401.

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

# A Appendix

## A.1 Preliminary

In this part, we present the mathematical notations employed in our theoretical inference. Specifically, the problem description and its associated notations are detailed in Table 2.

| Symbol | Description |
|---|---|
| $G$ | Directed graph |
| $V$ | Node set |
| $E$ | Edge set |
| $C_i$ | Group $i$ nodes in $V$ |
| $\mu$ | Edge weights |
| $u$ | Node activatation probability |
| $S$ | Seed set |
| $K$ | Maximum size of seed set |
| $I_G(S)$ | Expected number of influenced nodes by $S$ |
| $I_{G,C_i}(S)$ | Expected number of nodes in group $C_i$ influenced by seed set $S$ |
| $I_G(k)$ | Maximum number of nodes influence nodes under cardinality $k$ |
| $(\alpha,\beta)$-approximation | $Pr\left[r_\mu(S) \geq \alpha \cdot r_\mu(S^*)\right] \geq \beta$ |
| $E_S$ | Set of edges in the subgraph induced from $G$ by seed nodes $S$ |
| $r_\mu(S)$ | Optimization objective function with edge weights $\mu$ and seed set $S$ |

Table 2: Summary of notation used in problem description.

**Notation used in regret inference.** We begin by introducing the notations used in this paragraph. With approximation guarantee $(\alpha,\beta)$, we define seed nodes $S$ as a *bad set* if $r_\mu(S) < \alpha \cdot r_\mu(S^*)$, where $r_\mu(S^*)$ denotes the global optimal reward. The set of bad seed nodes is defined as $S_{\mathrm{B}} := \{S \mid r_\mu(S) < \alpha \cdot r_\mu(S^*)\}$. In CMAB, each edge in the graph represents the arm in the bandit. We let $S_{\mathrm{B},i} = \{S \in S_{\mathrm{B}} \mid i \in \tilde{E}_S\}$ denotes the set of bad seed nodes where the $i_{th}$ edge is expected to be reached. We order all bad seed nodes in $S_{\mathrm{B},i}$ as $S^1_{\mathrm{B},i}, S^2_{\mathrm{B},i}, \ldots, S^{K_i}_{\mathrm{B},i}$ in increasing order of their expected rewards, where $K_i = |S_{\mathrm{B},i}|$.

For any bad seed set $S \in S_{\mathrm{B}}$, with approximation guarantee $(\alpha,\beta)$, we define approximate regret $\Delta_S := \alpha \cdot r_\mu(S^*) - r_\mu(S)$. For the $i_{th}$ edge $i \in \{1,\ldots,|E|\}$ with $K_i > 0$ and reward order index $j \in \{1,\ldots,K_i\}$, we define $\Delta^{i,j} := \Delta_{S^j_{\mathrm{B},i}}$. Additionally, for convenience purpose, we let $\Delta^i_{\max} := \Delta^{i,1}$ and $\Delta^i_{\min} := \Delta^{i,K_i}$. Moreover, we define $\Delta_{\max} := \max_{i \in \{1,\ldots,|E|\}, K_i > 0} \Delta^i_{\max}$, and $\Delta_{\min} := \min_{i \in \{1,\ldots,|E|\}, K_i > 0} \Delta^i_{\min}$.

## A.2 Quoted Theorems

Our theoretical analysis rely on the following theorems from CUCB (Chen et al., 2014).

**Theorem 4** (Theorem 1 of CUCB (Chen et al., 2014)). *The $(\alpha,\beta)$-approximation regret of the CUCB algorithm in $T$ rounds using an $(\alpha,\beta)$-approximation oracle is at most*

$$\sum_{i \in \{1,\ldots,|E|\}, K_i > 0} \left( \ell_n\left(\Delta^i_{\min}, p_i\right) \Delta^i_{\min} + \int_{\Delta^i_{\min}}^{\Delta^i_{\max}} \ell_n\left(x, p_i\right) \mathrm{d}x \right) + \left(\frac{\pi^2}{2} + 1\right) \cdot |E| \cdot \Delta_{\max}, \tag{26}$$

*where $p^* = \min_{i \in \{1,\ldots,|E|\}} p_i$, and*

$$\ell_n(\Delta, p) = \max\left( \frac{12 \ln T}{(f^{-1}(\Delta))^2 p}, \frac{24 \ln T}{p} \right), 0 < p < 1 \tag{27}$$

**Theorem 5** (Theorem 2 of CUCB(Chen et al., 2014)). *Consider a CMAB problem with an $(\alpha,\beta)$-approximation oracle. Let $p^* = \min_{i \in \{1,\ldots,|E|\}} p_i$. If the bounded smoothness function $f(x) = \gamma \cdot x^\omega$ for some $\gamma > 0$ and $\omega > 0$, the regret of CUCB is at most*

$$\frac{2\gamma}{2-\omega} \cdot \left(\frac{12|E|\ln T}{p^*}\right)^{\omega/2} T^{1-\omega/2} + \left(\frac{\pi^2}{2} + 1\right) |E| \Delta_{\max} + \sum_{i \in \{1,\ldots,|E|\}} \frac{24 \ln T}{\mu_i} \Delta_{\max} \tag{28}$$

### A.3 Regret of Diversity Fairness

Here we provide a detailed proof for regret of diversity fairness in Equation (22).

For diversity fairness, the bounded smoothness function we use is $f(x) = |E||V|x + |V|$, and its corresponding inverse function is $f^{-1}(\Delta) = \left(\frac{\Delta - |V|}{|E||V|}\right)$.

Then, to determine $\ell_T(\Delta, p)$, our objective is to compare $\frac{1}{(f^{-1}(\Delta))^2}$ and 2 following $\ell_T(\Delta, p) = \max(\frac{12 \ln T}{(f^{-1}(\Delta))^2 p}, \frac{24 \ln T}{p})$ shown in Theorem 4. To make such a comparison, it is important to note $\Delta \leq |V|$. Consequently, $\frac{|E|^2 |V|^2}{(\Delta - |V|)^2}$ is evidently greater than 2. As a result, we can conclude that $\ell_T(\Delta, p) = \frac{12 \ln T |E|^2 |V|^2}{(\Delta - |V|)^2 p}$.

Finally, we provide the regret bound. By employing Theorem 4, we have the following regret formula:

$$\left(\frac{\pi^2}{2} + 1\right) \cdot |E| \cdot \Delta_{\max} \quad + \sum_{\substack{i \in \{1, \ldots, |E|\} \\ K_i > 0}} \left[ \frac{12 \ln T \cdot |E|^2 \cdot |V|^2 \cdot \Delta_{\min}^i}{\left(\Delta_{\min}^i - |V|\right)^2 p_i} \quad + \frac{12 \ln T \cdot |E|^2 \cdot |V|^2}{p_i} \left(\frac{1}{|V| - \Delta_{\max}^i} - \frac{1}{|V| - \Delta_{\min}^i}\right) \right] \tag{29}$$

Since $-\frac{1}{(|V| - \Delta_{min}^i)}$ is negative, we can remove it and obtain

$$\sum_{i \in E, K_i > 0} \left[ \frac{12 \ln T |E|^2 |V|^2}{(\Delta_{min}^i - |V|)^2 p_i} \Delta_{min}^i + \frac{12 \ln T |E|^2 |V|^2}{p_i (|V| - \Delta_{max}^i)} \right] + \left(\frac{\pi^2}{2} + 1\right) \cdot |E| \cdot \Delta_{\max} \tag{30}$$

### A.4 Regret of Maximin Fairness

In this section, we provide a detailed proof for the regret of maximin fairness in Equation (21).

For maximin fairness, we choose the bounded smoothness function as $f(x) = |E|x$.

Then, to determine $\ell_T(\Delta, p)$, we need to compare $\frac{1}{(f^{-1}(\Delta))^2}$ and 2. Because the objective function of maximin fairness is a proportion, we can naturally infer that $0 \leq \Delta \leq 1$. Consequently, we have $\frac{|E|^2}{\Delta^2} \geq \frac{|E|^2}{1} > 2$. And eventually, we get that $\ell_T(\Delta, p) = \frac{12 |E|^2 \ln T}{\Delta^2 \mu}$.

Finally, we show the regret bound. By employing Theorem 4, the regret upper bound is as follows.

$$\sum_{i \in \{1, \ldots, |E|\}, K_i > 0} \frac{24 \cdot |E|^2 \cdot \ln T}{\Delta_{\min}^i \cdot p_i} + \left(\frac{\pi^2}{2} + 1\right) \cdot |E| \cdot \Delta_{\max} \tag{31}$$

### A.5 Regret of Welfare Function

Here we provide the detailed proof of regret of welfare function in Section 5.1.

#### A.5.1 When $\alpha \geq 1$

Based on Equation (18), the bounded smoothness function we use is $f(x) = |E||V|x$.

To derive the upper bound for $f(x)$, we analyze $\Delta f(\Lambda, \alpha)$ using Lagrange median theorem. By transforming $\Delta f(\Lambda, \alpha)$ into difference form, which is $\Delta f(\Lambda, \alpha) = \sum_{c \in C} (|c|(\Lambda(\omega_c)^{\alpha - 1}))$ for $u_c \leq \omega_c \leq u_c + \Lambda$, where $\omega_c$ represents the corresponding gradient parameter used in Lagrange median theorem, we can bound it with $\Lambda$ and obtain the upper bound as $\Delta f(\Lambda, \alpha) = \sum_{c \in C} (|c| * (\Lambda))$. This implies that $f(x) = |E||V|x$ is a valid form of bounded smoothness function when $\alpha > 1$. Additionally, we aim to provide an upper bound for $f(x)$ in the form of $f(x) = \gamma x^\omega$. For $f(x) = |E||V| \frac{[(1+x)^\alpha - 1]}{\alpha}$, the upper bound bounded smoothness function is $f(x) = |E||V|(2^\alpha - 1)x^\alpha$. This result can be obtained by comparing the gradients and coincident starting points of the two functions.

In summary, we can have three forms of bounded smoothness function: $f(x) = |E||V|x$, $f(x) = |E||V|\frac{(1+x)^\alpha - 1}{\alpha}$ and $f(x) = |E||V|\frac{2^\alpha - 1}{\alpha}x^\alpha$.

To determine $\ell_T(\Delta, p)$, we use $f(x) = |E||V|\frac{(1+x)^\alpha - 1}{\alpha}$ and discuss it case by case.

Note that we have $\ell_T(\Delta, p) = \max(\frac{12 \ln T}{(f^{-1}(\Delta))^2 p}, \frac{24 \ln T}{p})$. Our main goal is to compare the relationship between $\frac{1}{(f^{-1}(\Delta))^2}$ and 2 with $f^{-1}(\Delta) = (\frac{\alpha \Delta}{|E||V|} + 1)^{\frac{1}{\alpha}} - 1$. Based on the above expression and definition, if $(\frac{\alpha \Delta}{|E||V|} + 1)^{\frac{1}{\alpha}} > 1 + \frac{1}{\sqrt{2}}$, we have $\ell_T(\Delta, p) = \frac{24 \ln T}{p}$. Otherwise, if $(\frac{\alpha \Delta}{|E||V|} + 1)^{\frac{1}{\alpha}} \leq 1 + \frac{1}{\sqrt{2}}$, we have $\ell_T(\Delta, p) = \frac{12 \ln T}{(f^{-1}(\Delta))^2 p}$.

$\ell_T(\Delta, p) = \frac{24 \ln T}{p}$ is mathematically impossible. If $\ell_T(\Delta, p) = \frac{24 \ln T}{p}$ happens to be true, we can re-formulate the requirement $(\frac{\alpha \Delta}{|E||V|} + 1)^{\frac{1}{\alpha}} \leq 1 + \frac{1}{\sqrt{2}}$ to $(1 + \frac{1}{\sqrt{2}})^\alpha < 1 + \frac{\Delta \alpha}{|E||V|}$. Given the magnitude of $|E|$ and $|V|$ assumed in the $G$ and the disparity between exponential and linear functions, the aforementioned inequality is almost impossible to be satisfied.

In conjunction with $\Delta_{max} = \frac{|V|}{\alpha}$ as the extreme case, solving $\frac{\alpha \Delta}{|E||V|} + 1 \leq (1 + \frac{1}{\sqrt{2}})^\alpha$ gives us $\alpha \geq \log_{1 + \frac{1}{\sqrt{2}}} 2$. Thus we have $\ell_T(\Delta, p) = \frac{12 \ln T}{(f^{-1}(\Delta))^2 p}$ if $\alpha \geq \log_{1 + \frac{1}{\sqrt{2}}} 2$.

Finally, we provide the problem-dependent regret bound. We use $\ell_T(\Delta, p)$ to derive the final regret result. In Theorem 4, for the sake of brevity, we use $f(x) = \gamma x^\omega$ to provide an upper bound for regret.

When $\alpha \geq \log_{1 + \frac{1}{\sqrt{2}}} 2$, we apply Theorem 4 with $f(x) = |E||V|(2^\alpha - 1)x^\alpha$. And the regret bound is applicable in the case where $\log_{1 + \frac{1}{\sqrt{2}}} 2 \leq \alpha$ and $\alpha \neq 2$

$$\sum_{\substack{i \in \{1, \ldots, |E|\} \\ K_i > 0}} \left[ \frac{12 \ln T \cdot \Delta_{min}^i}{\left[ \frac{\alpha}{(2^\alpha - 1)|E||V|} \Delta_{min}^i \right]^{\frac{2}{\alpha}} p_i} + \frac{12\alpha \ln T \left[ \left( \Delta_{max}^i \right)^{1 - \frac{2}{\alpha}} - \left( \Delta_{min}^i \right)^{1 - \frac{2}{\alpha}} \right]}{(\alpha - 2) p_i \left[ \frac{\alpha}{(2^\alpha - 1)|E||V|} \right]^{\frac{2}{\alpha}}} \right] + \left( \frac{\pi^2}{2} + 1 \right) |E|\Delta_{max}$$

If $\alpha = 2$, we simply apply Theorem 4 with $f(x) = 3|E||V|\frac{x^2}{2}$ and get

$$\sum_{i \in \{1, \ldots, |E|\}, K_i > 0} \left[ \frac{18|E||V|\ln T \cdot}{p_i} + \frac{36|E||V|\ln T}{p_i}(\Delta_{max}^i - \Delta_{min}^i) \right] + \left( \frac{\pi^2}{2} + 1 \right) |E|\Delta_{max} \tag{32}$$

In summary, if we take consideration of all cases, and notably include the fact that for $\alpha \geq 1$ there would always have $f(x) = |E||V|x$. In the case of $1 \leq \alpha \leq \log_{1 + \frac{1}{\sqrt{2}}} 2$, the regret upper bound would be same as original CUCB bound. We have the regret bound as follows.

- If $1 \leq \alpha \leq \log_{1 + \frac{1}{\sqrt{2}}} 2$, we have the regret as

$$\sum_{\substack{i \in \{1, \ldots, |E|\} \\ K_i > 0}} \frac{24 \cdot |V|^2 |E|^2 \ln T}{\Delta_{min}^i \cdot p_i} + \left( \frac{\pi^2}{2} + 1 \right) |E|\Delta_{max} \tag{33}$$

- If $\log_{1 + \frac{1}{\sqrt{2}}} 2 \leq \alpha \neq 2$, we have the regret as

$$\sum_{\substack{i \in \{1, \ldots, |E|\} \\ K_i > 0}} \left[ \frac{12 \ln T \cdot \Delta_{min}^i}{\left[ \frac{\alpha}{(2^\alpha - 1)|E||V|} \Delta_{min}^i \right]^{\frac{2}{\alpha}} p_i} + \frac{12\alpha \ln T \left[ (\Delta_{max}^i)^{1 - \frac{2}{\alpha}} - (\Delta_{min}^i)^{1 - \frac{2}{\alpha}} \right]}{(\alpha - 2)p_i \left[ \frac{\alpha}{(2^\alpha - 1)|E||V|} \right]^{\frac{2}{\alpha}}} \right] + \left( \frac{\pi^2}{2} + 1 \right) |E|\Delta_{max} \tag{34}$$

- If $\alpha = 2$, we have the regret as

$$\sum_{\substack{i \in \{1, \ldots, |E|\} \\ K_i > 0}} \left[ \frac{18|E||V|\ln T \cdot}{p_i} + \frac{36|E||V|\ln T(\Delta_{max}^i - \Delta_{min}^i)}{p_i} \right] + \left( \frac{\pi^2}{2} + 1 \right) |E|\Delta_{max} \tag{35}$$

### A.5.2  When $0 < \alpha \le 1$

In this case, we use the bounded smoothness function $f(x) = |E||V|\frac{x^\alpha}{\alpha}$.

To analyze $\ell_T(\Delta, p)$, our main goal is to compare the relationship between $\frac{1}{(f^{-1}(\Delta))^2}$ and 2, which entails comparing $(\frac{\alpha\Delta}{|E||V|})^{\frac{-2}{\alpha}}$ with 2. Since the upper bound of $\Delta$ is $\frac{|V|}{\alpha}$, if $\alpha \ge \frac{2\log|E|}{\log 2}$, then $\ell_T(\Delta, p) = \frac{24\ln T}{p}$. However, it is trivial that, $\alpha \ge \frac{2\log|E|}{\log 2}$ cannot be attained since $\alpha \in (0,1]$. Thus, for $\forall 0 < \alpha \le 1$, we have $\ell_T(\Delta, p) = \frac{12\ln T}{(f^{-1}(\Delta))^2 p}$.

Finally, the problem-dependent regret bound is

$$\sum_{\substack{i \in \{1, \dots, |E|\} \\ K_i > 0}} \left[ \frac{12\ln T \cdot \Delta_{min}^i}{\left[\frac{\alpha}{|E||V|}\Delta_{min}^i\right]^{\frac{2}{\alpha}} p_i} + \frac{12\alpha\ln T\left[(\Delta_{max}^i)^{1-\frac{2}{\alpha}} - (\Delta_{min}^i)^{1-\frac{2}{\alpha}}\right]}{(\alpha - 2)p_i\left[\frac{\alpha}{|E||V|}\right]^{\frac{2}{\alpha}}} \right] + \left(\frac{\pi^2}{2} + 1\right)|E|\Delta_{\max} \tag{36}$$

### A.5.3  When $\alpha < 0$

Specifically, in this case, we have $f(x) = |E||V|\frac{(1+\epsilon)^\alpha - \epsilon^\alpha}{\alpha} x^{-\alpha}$. The determination of an upper bound for $f(x)$ can be easily achieved by employing gradient and ensuring that the starting and ending points coincide. Eventually, we can have two $f(x)$, which respectively are $f(x) = |E||V|\frac{(1+\epsilon)^\alpha - \epsilon^\alpha}{\alpha} x^{-\alpha}$ and $f(x) = \frac{|E||V|((\epsilon+x)^\alpha - \epsilon^\alpha)}{\alpha}$.

Then we analyze $\ell_T(\Delta, p)$. Given that $f^{-1}(\Delta) = (\frac{\alpha\Delta}{|E||V|} + \epsilon^\alpha)^{\frac{1}{\alpha}} - \epsilon$ for the $f(x) = \frac{|E||V|((\epsilon+x)^\alpha - \epsilon^\alpha)}{\alpha}$, the objective would be to compare $\frac{1}{(f^{-1}(\Delta))^2}$ with 2. By solving the inequality, we know that $\frac{1}{2} > \left[(\frac{\alpha\Delta}{|E||V|} + \epsilon^\alpha)^{\frac{1}{\alpha}} - \epsilon\right]^2$ would result in $\ell_T(\Delta, p) = \frac{12\ln T}{(f^{-1}(\Delta))^2 p}$. To solve the aforementioned inequality, it is evident that $(\frac{\alpha\Delta}{|E||V|} + \epsilon^\alpha)^{\frac{1}{\alpha}} \ge \epsilon$. In the extreme scenario where $\Delta_{max} = \frac{|V|}{\alpha}$, the requirement for $\ell_T(\Delta, p) = \frac{12\ln T}{(f^{-1}(\Delta))^2 p}$ is $\frac{\sqrt{2}}{2} \ge (\frac{\alpha\Delta}{|E||V|} + \epsilon^\alpha)^{\frac{1}{\alpha}} - \epsilon$. Yet the condition $\frac{\sqrt{2}}{2} \le (\frac{\alpha\Delta}{|E||V|} + \epsilon^\alpha)^{\frac{1}{\alpha}} - \epsilon$ would result in $\ell_T(\Delta, p) = \frac{24\ln T}{p}$

Similar to previous cases, we now would have $\theta$ as the solution to the following equation.

$$\frac{1}{|E|} = (\frac{\sqrt{2}}{2} + \epsilon)^\theta - \epsilon^\theta \tag{37}$$

Hence, when $\alpha > \theta$, the expression for $\ell_T(\Delta, p)$ is given by $\frac{12\ln T}{(f^{-1}(\Delta))^2 p}$. Otherwise, when $\alpha < \theta$, it becomes $\ell_T(\Delta, p) = \frac{24\ln T}{p}$.

Finally, we provide the problem-dependent regret bound. As we proceed to use upper bounded smoothness function $f(x) = |E||V|\frac{(1+\epsilon)^\alpha - \epsilon^\alpha}{\alpha} x^{-\alpha}$ into Theorem 4, we obtain the following problem-dependent regrets:

By applying Theorem 4 with $f(x) = \gamma x^\omega$, we have the following problem-dependent regret bound

- If $\alpha \ge \theta$, we have the regret as

$$\sum_{\substack{i \in \{1, \dots, |E|\} \\ K_i > 0}} \left[ \frac{12\ln T \cdot \Delta_{min}^i}{\left[\frac{\alpha}{[(1+\epsilon)^\alpha - \epsilon^\alpha]|E||V|}\Delta_{min}^i\right]^{\frac{-2}{\alpha}} p_i} + \frac{12\alpha\ln T\left[(\Delta_{min}^i)^{1+\frac{2}{\alpha}}\right]}{(\alpha + 2)p_i\left[\frac{\alpha}{[(1+\epsilon)^\alpha - \epsilon^\alpha]|E||V|}\right]^{\frac{-2}{\alpha}}} \right] + \left(\frac{\pi^2}{2} + 1\right) \cdot |E| \cdot \Delta_{\max} \tag{38}$$

- If $\alpha \le \theta$, we have the regret as

$$\left(\frac{\pi^2}{2} + 1\right) \cdot |E| \cdot \Delta_{\max} + \sum_{i \in \{1, \dots, |E|\}} \frac{24\ln T}{p_i}\Delta_{max} \tag{39}$$

## A.6  Problem-Independent Regret

In this section, we outline the procedural steps to obtain the problem-independent regret bound for maximin fairness, diversity fairness, and welfare function. The focus of this section is to outline the procedural steps

involved in obtaining the problem-independent form of regret for Maximin Fairness, Diversity Fairness and Welfare Function.

### A.6.1 Problem-independent regret bound for maximin fairness

By utilizing the function $f(x) = |E|x$ and applying Theorem 5, we derive the problem-independent form of regret as follows

$$\sqrt{\frac{48|E|^3 T \ln T}{p^*}} + \left(\frac{\pi^2}{2} + 1\right) \cdot |E| \cdot \Delta_{\max} \tag{40}$$

### A.6.2 Problem-independent regret bound for diversity fairness

Since in previous works, bounded smoothness and Theorem5 under the assumption that $f(0) = 0$. We need to introduce the justification of assumption $f(0) \neq 0$ in bounded smoothness can still be employed to give problem-independent regret guarantee. So we modify the proof steps in CUCB of Theorem 5 to give a modified problem-independent regret upper bound. The result of such modification would be that we have an additional $2cT$ regret bound, where $f(x) = c + \gamma x^\omega$.

Thus, for $f(x) = |V| + |E||V|x$, we can have the following regret bound

$$|V|\sqrt{\frac{48|E|^3 T \ln T}{p^*}} + \left(\frac{\pi^2}{2} + 1\right) \cdot |E| \cdot \Delta_{\max} + 2|V|T \tag{41}$$

### A.6.3 Problem-independent regret bound for welfare function

Here we respectively discuss the problem-independent regret in regard to the value of $\alpha$ and $\ell_T(\Delta, p)$.

**When $\alpha \geq 1$:** The value of $\ell_T(\Delta, p)$ can be seen in Appendix A.5.1 and use $f(x)$ in form of $f(x) = \gamma x^\omega$, which is $f(x) = |E||V|(2^\alpha - 1)x^\alpha$. Then by utilizing Theorem5, we can have problem-independent regret as follows.

- If $1 \leq \alpha \leq \log_{1 + \frac{1}{\sqrt{2}}} 2$ or $\alpha = 2$, we have the problem-independent regret as

$$|V|\sqrt{\frac{48|E|^3 T \ln T}{p^*}} + \left(\frac{\pi^2}{2} + 1\right) \cdot |E| \cdot \Delta_{\max} \tag{42}$$

- If $\log_{1 + \frac{1}{\sqrt{2}}} 2 \leq \alpha \neq 2$, we have the problem-independent regret as

$$\frac{2}{2 - \alpha} \frac{|E||V|}{\alpha} (2^\alpha - 1) [\frac{12|E| \ln(V)}{p^*}]^{\frac{\alpha}{2}} |V|^{1 - \frac{\alpha}{2}} + \left(\frac{\pi^2}{2} + 1\right) |E| \Delta_{\max} \tag{43}$$

**When $0 < \alpha \leq 1$:** The value of $\ell_T(\Delta, p)$ can be seen in Appendix A.5.2, and we can simply use $f(x) = |E||V|\frac{x^\alpha}{\alpha}$ for problem-independent bound derived from Theorem 5. Thus the problem-independent regret would be

$$\frac{2}{2 - \alpha} \frac{|E||V|}{\alpha} [\frac{12|E| \ln(V)}{p^*}]^{\frac{\alpha}{2}} |V|^{1 - \frac{\alpha}{2}} + \left(\frac{\pi^2}{2} + 1\right) |E| \Delta_{\max} \tag{44}$$

**When $\alpha < 0$:** The $f(x)$ here would be $f(x) = |E||V|\frac{(1+\epsilon)^\alpha - \epsilon^\alpha}{\alpha} x^{-\alpha}$. Thus, by applying Theorem 5, we have the following regret upper bound under the condition that $\alpha \geq \theta$.

$$\frac{2}{2 + \alpha} \cdot \frac{|E||V|[(1 + \epsilon)^\alpha - \epsilon^\alpha]}{\alpha} \cdot \left[\frac{12|E| \ln(V)}{p^*}\right]^{\frac{-\alpha}{2}} \cdot |V|^{1 + \frac{\alpha}{2}} + \left(\frac{\pi^2}{2} + 1\right) \cdot |E| \cdot \Delta_{\max} \tag{45}$$

In addition, we would like to point out that, for $\alpha \leq \theta$, we have that $\ell_T(\Delta, p) = \frac{24 \ln n}{p}$. With the different $\ell_T(\Delta, p)$ form, the problem-independent regret in Theorem 5 would be employed without the $\frac{2\gamma}{2-\omega} \cdot \left( \frac{12|E|\ln T}{p^*} \right)^{\omega/2} T^{1-\omega/2}$ part, resulting in upper regret bound same to problem-dependent case, i.e.,

$$\left( \frac{\pi^2}{2} + 1 \right) \cdot |E| \cdot \Delta_{\max} + \sum_{i \in \{1, \ldots, |E|\}} \frac{24 \ln T}{p_i} \Delta_{max} \tag{46}$$

### A.7 Dataset Descriptions

In our experiments, we use various widely-used benchmark datasets for fair graph learning: NBA, German, Pokec-z, bail and Youtube. These datasets are common datasets used in in fair graph learning. For example, NBA dataset was used in FairGNN (Dai & Wang, 2020) with Nationality as sensitive attribute and Bail dataset was used in GEAR (Ma et al., 2022) with Race as sensitive attribute. We use the same sensitive attribute as in references. Attributes like (Race, etc) are sensitive in real scenarios when considering fair online influence maximization. The detailed dataset descriptions are as follows.

- NBA is a social network of NBA players on Twitter. Two players (nodes) are connected if one follow another on Twitter. We use the nationality of each user as the sensitive attribute.

- German is a similarity graph of clients in a Germany bank. Nodes are the clients, and two nodes are connected if their credit accounts are similar. We use the age of each client as the sensitive attribute.

- Pokec-z is a sub-network of the Slovakian social network *Pokec* of one province. Each node is a user in the selected province, and edges represent the friendship relationship among users. We use the biological sex of each user as the sensitive attribute.

- YouTube is a social network of user friendship relationship on the video-sharing web platform YouTube. Each node is a user on YouTube, and two nodes are connected if they form the friendship with each other. We use the community membership of each user as the sensitive attribute.

### A.8 Additional Experiments

In this section, we provide evidence as additional support for our Experiments part.

#### A.8.1 Comparison between different submodular oracles

We compare the performance of FOIM with two submodular oracles: degree discount and greedy strategy. From Figure 5, we can conclude that the two compared submodular oracles have many similar regret. Thus, in our experiments in the main body (i.e., Figure 3 and Figure 2), we only use degree discount.

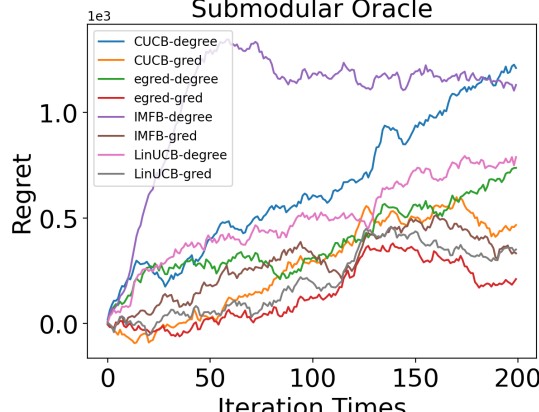

Figure 5: Cumulative regrets using submodular oracles in NBA. Methods with suffix -degree use use degree discount (Chen et al., 2009) as the offline oracle, while methods with suffix -greedy use greedy selection as the offline oracle.

#### A.8.2 Experiments on German

We provide additional experimental results on German dataset, as shown in Figure 6. From the figure, we observe that the IMLinUCB has poor performance in German, as the regret of IMLinUCB is much larger. This might be due to a few reasons. First, for a dense network like German, the seed size we select is relatively large. Second, the sensitive attribute, i.e., age, contains more possible sensitive attribute values (non-binary), making the objective function insensitive to context, which is critical in IMLinUCB. Other than IMLinUCB, we can still observe the superiority of fair oracle under maximin regret in German.

### A.8.3   Experiments on bail dataset

We present additional experimental results on bail dataset, as shown in Figure 7. We employed the MIP as fair oracle and compared it with the results of baselines, and this figure can serve as an additional proof of the superiority of fair oracle when the reward is fair.

### A.8.4   Scalability demonstration

To demonstrate our algorithm's scalability to large graph dataset spanning million of nodes, we tested FOIM on Youtube dataset with MIP as fair oracle and compared with the baselines. The results can be seem in Figure 8 and can serve as additional proof of the performance of FOIM framework. Due to the size of dataset and computation resource constraint, we only ran on it one time and thus there is no deviation.

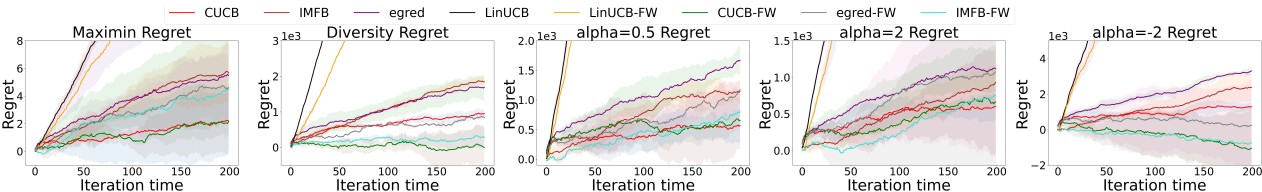

Figure 6: Cumulative regrets under maximin fairness, diversity fairness, and welfare function ($\alpha = -2, 2, 0.5$) in german. Methods with suffix -FW use the multi-objective Frank Wolfe algorithm (Tsang et al., 2019) as offline oracle with fairness consideration. Methods without suffix use degree discount (Chen et al., 2009) as the offline oracle. Shaded area refers to the standard deviation of the corresponding method across different runs.

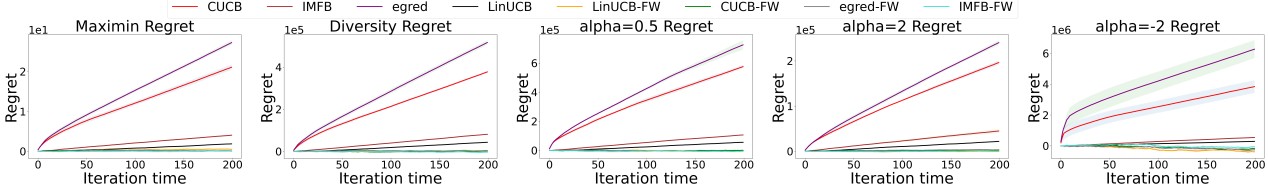

Figure 7: Cumulative regrets under maximin fairness, diversity fairness, and welfare function ($\alpha = -2, 2, 0.5$) in bail. Methods with suffix -MIP use the Mixed Integer Programming algorithm (Farnadi et al., 2020) as offline oracle with fairness consideration. Methods without suffix use degree discount (Chen et al., 2009) as the offline oracle. Shaded area refers to the standard deviation of the corresponding method across different runs.

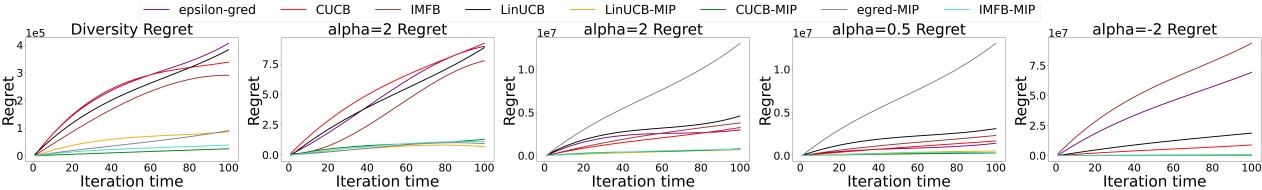

Figure 8: Cumulative regrets under maximin fairness, diversity fairness, and welfare function ($\alpha = -2, 2, 0.5$) in Youtube dataset. Methods with suffix -MIP use the Mixed Integer Programming algorithm (Farnadi et al., 2020) as offline oracle with fairness consideration. Methods without suffix use degree discount (Chen et al., 2009) as the offline oracle.

### A.8.5 Regret between FOIM and offline fair oracle

We evaluate the regret performance of a Frank-Wolfe offline oracle on the NBA dataset using the diversity fairness metric, under perturbed activation probabilities. Specifically, Gaussian noise sampled from $\mathcal{N}(0, 0.05^2)$ is added to the base activation probabilities to simulate realistic imperfections in offline estimates. Dotted line in Figure 9 presents a generally higher cumulative regret.

This setup highlights the sensitivity of offline methods to noise in activation estimates. In contrast, the adaptive online methods dynamically adjust to the true graph structure and exhibit lower expected regret under the same noise conditions.

### A.8.6 Experiment of comparison between offline fair algorithms and online fair algorithms.

To support the claim in Introduction part "Consequently, it is likely that the estimated probabilities are inconsistent with respect to individuals in the minority group due to bias introduced in the offline data selection (e.g., less logged activations for individuals from the protected group)", we performed a new experiment on pokec dataset with "Sex" as sensitive attribute and select the group with labeled as Sex="1". We set seed size as 300 to observe the performance of influenced nodes in the last round of online learning (100 iterations) and compare it to the output of offline oracle (MIP algorithm). We evaluate the results on the metric of estimation error, which is the absolute of the gap between estimated propagation probability and ground truth probability. We need to note that the estimation error in offline algorithm is more

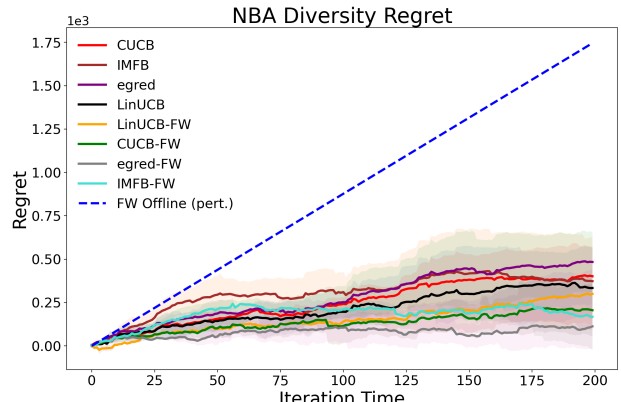

Figure 9: Diversity regret on NBA. Dashed line: FW-Offline with $\mathcal{N}(0, 0.05^2)$.

related to offline data selection, which is like a data sampling process and somehow independent of the operating of Frank-Wolfe algorithm and other fair oracles like MIP. For example, on MIP, the offline data selection is achieved by Monte Carlo Sampling. To report and ensure the reliability of our evaluation, we report Table 3 with standard deviation after 5 parallel executions.

|  | MIP(offline) | CUCB | IMFB | IMLinUCB |
|---|---|---|---|---|
| Estimation error | $0.219 \pm 0.047$ | $0.177 \pm 0.055$ | $0.199 \pm 0.022$ | $0.151 \pm 0.050$ |
| Influenced nodes | $1488 \pm 14.2$ | $1120 \pm 10.7$ | $1507 \pm 18.8$ | $1525 \pm 15.0$ |

Table 3: Comparison between online fair algorithms and offline fair algorithms

We record the comparison in the Table 3. The experiment indicates that the final seed selection of fair online learning might be better than offline fair algorithm and supports the statement "It may further result in sub-optimal seed selection for fair offline influence maximization that relies on the estimated information.".

### A.8.7 Influence spread and fairness trade-off

In addition to the online and offline fair algorithm comparison, we also report new results of influence spread of fair algorithms and baselines at Table 4 measured by the number of activated nodes at the last round ($T = 200$) on the Pokec-Z dataset, which includes 6,185 nodes in total. Compared to their non-fair counterparts that only maximize influence, we observe that fair algorithms by FOIM achieve similar or smaller influence spread under different metrics. Specifically, influence spread is almost not affected by diversity fairness. For maximin fairness and welfare function, the spread is $\sim 30$ smaller than non-fair baselines with IMLinUCB and IMFB, two efficient algorithms that leverage/estimate features of nodes and edges. The reduced influence spread under fairness constraints is expected and is the price of trading-off between fairness and influence.

Table 4: Influence spread and fairness trade-off on the Pokec-Z dataset.

| Influence | Maximin fairness | Diversity fairness | Welfare ($\alpha = 0.5$) | $\alpha = 2$ | $\alpha = -2$ |
|---|---|---|---|---|---|
| CUCB | 1289 | 1298 | 1243 | 1221 | 1251 |
| epsilon-greedy | 1246 | 1234 | 1238 | 1237 | 1217 |
| IMFB | 1481 | 1437 | 1483 | 1460 | 1471 |
| IMLinUCB | 1387 | 1334 | 1419 | 1338 | 1409 |
| CUCB-FW | 1038 | 1324 | 1220 | 1163 | 1132 |
| epsilon-greedy-FW | 974 | 1372 | 1259 | 1056 | 1190 |
| IMFB-FW | 1110 | 1431 | 1177 | 1192 | 1192 |
| IMLinUCB-FW | 1074 | 1383 | 1188 | 1074 | 1116 |

## A.9 Additional Proof of Diversity Fairness

In this section, we provide the additional proof of diversity fairness that if the diversity constraint is satisfied in $G$ with utility vector as $\mu$, the constraint will not be violated for $G'$ where $\mu'_i = \mu_i + \Lambda$.

*Proof.* **If the diversity constraint is satisfied with $\mu$, the constraint will not be violated after the lifting of $\mu$ (adding $\Lambda$ to $\mu$)**

If we only investigate on an edge $j$ and let $\mu'_j = \mu_j + \Lambda$, the increase of $I_{G,C_i}(S)$ is larger than $I_{G[C_i]}(k_i)$ would complete the proof, as the general proof of lifting $\mu$ is just a combination of each increase to edge propagation weight.

Thus, we give the formal proof of setting $\mu'_j = \mu_j + \Lambda$ to have the increase of $I_{G,C_i}(S)$ is larger than $I_{G[C_i]}(k_i)$.

**Case 1.** If edge $\mu_j$ is out of $G[C_i]$, the proof naturally stands. We can notice that $I_{G[C_i]}(k_i)$ is the maximal expectation that fair allocation $k_i$ can achieve on the deduced subgraph $G[C_i]$, which is independent of edge $\mu_j$ out of the subgraph. Since we have condition that $I_{G,C_i}(S) \geq I_{G[C_i]}(k_i)$ before the lifting(adding $\Lambda$) process, yet adding edge $\mu_j$ is out of $G[C_i]$ would only possibly increase on $I_{G,C_i}(S)$, completing the proof in this case.

**Case 2.** If edge $\mu_j$ is within $G[C_i]$, the proof goes as the followings. We consider the case that we lift edge $\mu_j$ to $\mu'_j$ and the increase of spread to be $\Lambda$, and we mark the end node of $\mu_j$ as $y$. Thus, the increase of $I_{G,C_i}(S)$ and $I_{G[C_i]}(k_i)$ is determined by the increased spread starting from $y$. The the increase of $I_{G[C_i]}(k_i)$ would be at most $\Lambda I_{G'[C_i]}(y)$ and the increase of $I_{G,C_i}(S)$ would be $\Lambda I_{G',C_i}(y)$ at most. However, the increase would not always reach the maximal value like $\Lambda I_{G'[C_i]}(y)$ and $\Lambda I_{G',C_i}(y)$. Since the spread from $y$ is not on an empty graph, the nodes in $C_i$ may already be activated. For example, $\Lambda I_{G'[C_i]}(y)$ would have non-zero influence to its neighbors, yet the neighbors already have activation probability of 1.0 as they are seed nodes.

**2.1** We first discuss about when $\Lambda I_{G'[C_i]}(y)$ and $\Lambda I_{G',C_i}(y)$ are the increases. We look at the volume relationship between $\Lambda I_{G'[C_i]}(y)$ and $\Lambda I_{G',C_i}(y)$. As $G'[C_i]$ is subgraph of $G'$, implying that the increase of $I_{G'[C_i]}(y)$ is also made $I_{G',C_i}(y)$ in $C_i$, so $I_{G',C_i}(y)$ would only be larger or equal than $I_{G'[C_i]}(y)$ in $C_i$ as it would spread to groups other than $C_i$ and may propagate back to $C_i$, making the influence strictly larger.

**2.2** We analyze the case when increase starting from $y$ in each metric is smaller than $\Lambda I_{G'[C_i]}(y)$ and $\Lambda I_{G',C_i}(y)$ in this part.

This case would happen as the increase spreading from $y$ is not on an empty graph, and each node in $C_i$ has already has it's expected activation probability due to the $I_{G'[C_i]}(S)$ or $I_{G',C_i}(S)$. For example, if we employ $\Lambda I_{G'[C_i]}(y)$ as increase, and $\Lambda I_{G'[C_i]}(y)$ would have allocated increase in each node in $C_i$. Yet there exists such constraint that for each node in $C_i$, the allocated increase plus it's original expected activation probability must be smaller than 1, explaining why this case happen. As in some nodes, due to the existing of the constraint, the increase could only be smaller than the allocation of spread from $y$.

For each node $o$ in class $C_i$, we have the following notations. $P_o$ is the what $I_{G'[C_i]}(y)$ has on node $o$, and $R_o$ is the allocation of $I_{G[C_i]}(k_i)$ on node $o$, $M_o$ is the allocation of $I_{G',C_i}(y)$, and $B_o$ of $I_{G,C_i}(S)$. They represent

what the activation probability of each node in different diffusion process. For example $I_{G'[C_i]}(y)$ can be expressed as $\sum_{o \in C_i} P_o$, and $R_o = 1$ or $B_o = 1$ represents $o$ is seed node.

$$Gain_o = \begin{cases} \Lambda P_o & \forall o \in C_i, 1 - \Lambda P_o - R_o \geq 0 \\ 1 - R_o & \forall o \in C_i, 1 - \Lambda P_o - R_o \leq 0 \end{cases} \tag{47}$$

Due to the existence of the constraint in Equation 47, we can divide the nodes in $C_i$ into two groups respectively in considering $I_{G[C_i]}(S)$ and $I_{G,C_i}(S)$. We annotate node set $\Omega$ as $\{o \in C_i, \Lambda P_o \leq 1 - R_o\}$, node set $\Theta$ as $\{o \in C_i, \Lambda M_o \leq 1 - B_o\}$. For node o in $\Omega$ or $\Theta$, the increase would be $\Lambda P_o$ and $\Lambda M_o$. Yet for node $o$ in $C_i - \Omega$ or $C_i - \Theta$, the increase would be $1 - R_o$ and $1 - B_o$.

We compare the increase of spread in $C_i$ under the constraint. For increase under $I_{G[C_i]}(k_i)$, the general spread would be $\sum_{o \in \Omega} \Lambda P_o + \sum_{o \in C_i - \Omega}(1 - R_o)$. And for the increase of $I_{G,C_i}(S)$ would be $\sum_{o \in \Theta} \Lambda M_o + \sum_{o \in C_i - \Theta}(1 - B_o)$ following the same logic. We target to compare the relationship between the both constrained increase.

Since we already have relationship of $I_{G',C_i}(y)$ and $I_{G'[C_i]}(y)$, $I_{G'[C_i]}(k_i)$ and $I_{G',C_i}(S)$ in Case2 and our pre-condition, and the conditions can be expressed as $\Lambda \sum_{o \in C_i} P_o \leq \Lambda \sum_{o \in C_i} M_o$, $\sum_{o \in C_i} R_o \leq \sum_{o \in C_i} B_o$. Combining these two conditions with Equation 47, we can have

Hereby we target to have $\sum_{o \in C_i}(1 - R_o) + \sum_{o \in \Omega}(\Lambda P_o + R_o - 1)$ minused by $\sum_{o \in C_i}(1 - B_o) + \sum_{o \in \Theta}(\Lambda M_o + B_o - 1)$ and compare it to 0. After calculation, we can have the gap expressed as $\sum_{o \in C_i}(\Lambda M_o - \Lambda P_o) - \sum_{o \in C_i - \Omega}(\Lambda P_o + R_o - 1) + \sum_{o \in C_i - \Theta}(\Lambda M_o + B_o - 1)$. Since only in set $C_i - \Theta$, $\Lambda M_o + B_o - 1$ can be positive and reach the maximal value, which means we can have $\sum_{o \in C_i - \Omega}(\Lambda M_o + B_o - 1) \leq \sum_{o \in C_i - \Theta}(\Lambda M_o + B_o - 1)$. Thus we can have the final result represented as $\sum_{o \in C_i}(\Lambda M_o - \Lambda P_o) - \sum_{o \in C_i - \Omega}(-\Lambda P_o - R_o + \Lambda M_o + B_o)$. Taking the conditions and constraints into consideration, the final result would be larger than 0.

$\square$

## A.10 Bounded Smoothness for Diversity Fairness

*Proof.* We consider two cases here.

(1) Given feasible seed nodes $S$, both $G$ and $G'$ satisfy diversity constraint. In this case, the problem is equivalent to influence maximization without fairness constraint with reward function being Thus, it satisfies bounded smoothness with function $f(x) = |E||V|x$ (Chen et al., 2014).

(2) Given feasible seed nodes $S$, at least one of $G$ and $G'$ violates the diversity constraint. Similarly, by Lemma 3, it is impossible that $G$ satisfies the diversity constraint while $G'$ violates. Then

- If both $G$ and $G'$ violate the diversity constraint, we have that $r_\mu^{diversity}(S) = r_{\mu'}^{diversity}(S) = 0$.
- If $G$ violates the diversity constraint while $G'$ satisfies, the largest reward increase is achieved by replacing $\mu_i$ with $\mu_i'$ in $G$ $\forall i \in \{1, ..., |E'|\}$, which will make $r_\mu^{diversity} > 0$. Given that the diversity constraint is already satisfied in the graph, the increase of the reward brought by edge $i$ in $\tilde{E}_S$ is at most $|V|\Lambda$. If we regard the reward increase brought by $\tilde{E}_S$ as a sequential process, i.e., the reward increase is brought by each edge in $\tilde{E}_S$ sequentially, then the total increase of the reward brought by $\tilde{E}_S$ in $G'$ must be smaller than $(s - 1)|V|\Lambda + h$, where $h$ denotes the reward increase bought by the first edge making $r_\mu^{diversity} \neq 0$. Since $h \leq |V|$, we have $|r_{\mu'}^{diversity}(S) - r_\mu^{diversity}(S)| \leq (s - 1)|V|\Lambda + |V| \leq |E||V|\Lambda + |V|$.

Combining (1) and (2), we complete the proof for bounded smoothness with smoothness function $f(x) = |E||V|x + |V|$. $\square$

## A.11 Detailed Steps of Bounded Smoothness of Welfare Function

In this section, a comprehensive analysis is presented, elucidating the upper bound derivation for the expression of bound function, $|c|\frac{(u^c + \Lambda)^\alpha - (u^c)^\alpha}{\alpha}$ within the context of the welfare function, associated with value of $\alpha$. Given $\alpha$ and $\Lambda$, we define $g(u^c) = |c|\frac{(u^c + \Lambda)^\alpha - (u^c)^\alpha}{\alpha}$. Therefore, the maximum increase of the welfare reward

function bought by any one edge equals to $\sum_{c \in C} g(u^c)$. Here we take the derivative of $g$ with respect to any $u^c$. We have $\frac{\partial g}{\partial u^c} = (u^c + \Lambda)^{\alpha-1} - (u^c)^{\alpha-1}$.

(1) If $\alpha \geq 1$: Since $0 < u^c \leq 1$, we have $\frac{\partial g}{\partial u^c} \geq 0$. Therefore, $g(u^c)$ is strictly increasing regard to $u^c$. In this case, we have the upper bound for the increase of the welfare reward function bring by any edge $\sum_{c \in C}(|c|(1+\Lambda)^{\alpha})/\alpha - \sum_{c \in C}(|c|)/\alpha$

(2) If $0 < \alpha < 1$: we have $\frac{\partial g}{\partial u^c} < 0$. Therefore, $g(u^c)$ is strictly decreasing with regard to $u^c$. Consequently, we have the upper bound for the increase of the welfare reward function bring by any edge $\sum_{c \in C}(|c|(\Lambda)^{\alpha})/\alpha$.

(3) If $\alpha < 0$: we let $\epsilon = \min(\{u^c | \forall c \in C\})$. Since $\forall c \in C$, $u^c \geq 0$, we have $\epsilon \geq 0$. Since $\frac{\partial g}{\partial u^c} < 0$, the welfare reward function achieves the largest vale under the condition that $\forall c \in C$, $u^c = \epsilon$. Therefore, the upper bound equals to the welfare IM reward increase when $\forall c \in C$, $u^c = \epsilon$, i.e., $\sum_{c \in C}(|c|(\epsilon + \Lambda)^{\alpha})/\alpha - \sum_{c \in C}(|c|\epsilon^{\alpha})/\alpha$.

### A.12 Time Complexity Analysis

In this part, we delve into the comparison between FOIM algorithms and original online IM algorithms. Notably, FOIM also operates within the CMAB framework, akin to the foundational online algorithms.

Taking the example of the CUCB algorithm, its time complexity is expressed as $O(T(|E|\log|E|+ORACLE_{\text{time}}))$ (Chen et al., 2014). Consequently, the bottleneck of our analysis hinges on the computational cost associated with the ORACLE, denoted as $ORACLE_{\text{time}}$. Intuitively, the time complexity tends to be higher for fair oracles compared to unfair ones, due to trade-off between efficiency and fairness.

The time complexity would be $O(\frac{(|E|+m)^2 k^2 b}{\epsilon} \log^2 \frac{|V|}{\delta})$ (Tsang et al., 2019) for of Frank-Wolfe fair oracle, where $m$ is the number of demographic groups, $k$ is the number of seed nodes, $b = \max_{i,j}(\frac{I_{G,C_i}(j)}{|C_i|})$ is the maximum normalized influence to a demographic group, $\epsilon$ is an error residual, and $\delta$ is the probability threshold where $\epsilon$ and $\delta$ exists in the $(\alpha, \beta)$ approximation (Tsang et al., 2019). However, the time complexity for unfair oracle like general greedy and degree discount is $\Omega(m|E|k \cdot POLY(\epsilon^{-1}))$ (Borgs et al., 2012; Kempe et al., 2003).

In addition, at the Table 5 we report the average running time of our algorithm (with suffix -FW) and non-fair algorithms (without suffix) in the table above. The Youtube dataset (the largest dataset) is run on Google Cloud with N4 VM and Google Axion Processor including 32 vCPUs and 128 GB of DDR5 memory, while other datasets are tested on an AMD 5800H CPU. The non-fair methods (without suffix) use the efficient degree discount oracle, while the fair algorithms use the multi-objective Frank-Wolfe as a fair oracle.

From the table, we observe that FOIM algorithms tend to have a larger running time but are within the acceptable range. For example, on the largest dataset, Youtube, fair algorithms are five to six times slower than the non-fair baseline. The additional running time is because the fair ORACLE is slower to solve the optimization problem and could be further reduced given more efficient fair offline IM solvers in the future.

Table 5: Average running time of algorithms on various datasets.

| Algorithm | NBA | Pokec-Z | German | Bail | Youtube |
|---|---|---|---|---|---|
| CUCB | 0:00:15 | 0:01:14 | 0:00:57 | 0:08:48 | 0:33:41 |
| epsilon-greedy | 0:00:12 | 0:01:01 | 0:00:46 | 0:07:31 | 0:30:18 |
| IMFB | 0:00:38 | 0:02:13 | 0:02:20 | 0:14:48 | 0:59:49 |
| IMLinUCB | 0:00:26 | 0:02:03 | 0:04:53 | 0:12:57 | 0:47:48 |
| CUCB-FW | 0:02:32 | 0:12:47 | 0:06:21 | 1:03:45 | 3:46:39 |
| epsilon-greedy-FW | 0:02:24 | 0:12:10 | 0:06:07 | 0:53:00 | 3:30:51 |
| IMFB-FW | 0:03:06 | 0:13:59 | 0:08:53 | 1:38:29 | 4:18:06 |
| IMLinUCB-FW | 0:02:57 | 0:13:43 | 0:08:05 | 1:27:26 | 4:03:00 |

### A.13 Discussion of TPM Condition

In the context of OIM SOTA algorithms, the Tightening Probability Mass (TPM) condition Wang & Chen (2017b); Wen et al. (2016) provides a smoothness assumption that is weaker and more broadly applicable than the classical bounded smoothness condition. We investigate whether common fairness-aware objective functions satisfy this condition. In particular, we compare the behavior of Diversity Fairness and Maximin Fairness under TPM.

**Theorem 6** (TPM Condition). *Let $\mu$ and $\mu'$ be two influence probability vectors over the edge set, and let $S$ be a seed set. A reward function $r_\mu(S)$ satisfies the* Tightening Probability Mass (TPM) *condition if there exists a constant $B > 0$ such that*

$$|r_\mu(S) - r_{\mu'}(S)| \leq B \sum_i p_i^S |\mu_i - \mu_i'|, \tag{48}$$

*where $p_i^S$ is the probability that edge $i$ is activated (i.e., traversed) under the influence process given seed set $S$.*

**Diversity Fairness Violates the TPM Condition.**

*Proof.* Assume a seed set $S$ marginally satisfies the diversity constraint under edge probabilities $\mu$, resulting in a non-zero reward:

$$r_\mu(S) > 0. \tag{49}$$

Now perturb a single edge probability by a small amount $\epsilon$, yielding a new probability vector $\mu'$. This small change may violate the diversity constraint, causing the reward to drop sharply:

$$r_{\mu'}(S) = 0. \tag{50}$$

Thus, the reward difference is a constant:

$$|r_\mu(S) - r_{\mu'}(S)| = c > 0, \tag{51}$$

whereas the TPM term vanishes as $\epsilon \to 0$:

$$\sum_i p_i^S |\mu_i - \mu_i'| \to 0. \tag{52}$$

Therefore, there exists no constant $B$ such that

$$|r_\mu(S) - r_{\mu'}(S)| \leq B \sum_i p_i^S |\mu_i - \mu_i'|, \tag{53}$$

implying that the TPM condition fails for any $B$. Hence, the diversity fairness objective does not satisfy the TPM condition. $\qquad\square$

**Maximin Fairness Satisfies the TPM Condition.**

*Proof.* Define the maximin fairness reward as

$$r_\mu^{\mathrm{maximin}}(S) = \min_{C \in \mathcal{C}} \frac{IG_C(S, \mu)}{|C|}, \tag{54}$$

where $IG_C(S, \mu)$ denotes the expected influence on group $C$. Let $\mu'$ be a perturbed influence probability vector.

For any group $C$, we have:

$$\left| \frac{IG_C(S, \mu)}{|C|} - \frac{IG_C(S, \mu')}{|C|} \right| = \frac{1}{|C|} |IG_C(S, \mu) - IG_C(S, \mu')|. \tag{55}$$

Assuming the Independent Cascade (IC) model and Lipschitz continuity of the influence function over edge weights:

$$|IG_C(S,\mu) - IG_C(S,\mu')| \leq \sum_{i \in E_S} p_i^S |\mu_i - \mu_i'|, \tag{56}$$

where $E_S$ is the set of edges that may be activated under seed set $S$.

It follows that:

$$\left| \frac{IG_C(S,\mu)}{|C|} - \frac{IG_C(S,\mu')}{|C|} \right| \leq \frac{1}{|C|} \sum_{i \in E_S} p_i^S |\mu_i - \mu_i'|. \tag{57}$$

Taking the worst-case deviation over all groups:

$$|r_\mu^{\text{maxmin}}(S) - r_{\mu'}^{\text{maxmin}}(S)| \leq \max_{C \in \mathcal{C}} \frac{1}{|C|} \sum_{i \in E_S} p_i^S |\mu_i - \mu_i'|. \tag{58}$$

Letting

$$B = \max_{C \in \mathcal{C}} \frac{1}{|C|}, \tag{59}$$

we finally obtain:

$$|r_\mu^{\text{maxmin}}(S) - r_{\mu'}^{\text{maxmin}}(S)| \leq B \sum_{i \in E_S} p_i^S |\mu_i - \mu_i'|, \tag{60}$$

which satisfies the TPM condition. $\qquad \square$

