# OpenReview forum: "Fair Online Influence Maximization"
_TMLR — Accepted by TMLR_

### Review · Reviewer_q7Vx · 2024-12-18

**Summary Of Contributions:**

This paper has introduced the first **Fair Online Influence Maximization (FOIM)** framework which considers online learning for Fair Influence Maximization problems taking into account group fairness perspective.


The FOIM framework combines ideas from :
  - online Influence Maximization and
  - Fair Influence Maximization.

The framework is general in a sense that it can combine many Online Influence Maximization approaches with Many fairness constraints which are the ones that satisfy the two mentioned properties of Monotonicity and Bounded smoothness.

This paper investigates three different existing fairness constraints to define rewards and analyzes the regret bound of combining them with  an instance of combinatorial multi-armed bandit approach (CMAB).

And finally, they have empirically shown that these combinations outperforms the online influence maximization algorithms that do not take fairness into account. Moreover, they have shown that using fair oracle would lead to similar estimation of the graph weights compared to submodular oracles.

**Audience:**

Yes

**Broader Impact Concerns:**

Based on my current understanding, I do not think this work needs a broader impact concern section.

**Claims And Evidence:**

No

**Requested Changes:**

1. The paper needs to improve on the weakness points I made earlier in **Strengths and Weaknesses** box.
2. There are some typos or notation abuse in the paper:
      - Lemma 3, P1: $i$ is used both for each edge and enumeration of the community $C_i$, in other words, does $C_i$ refer to the $i^{th}$ community or the community that includes $i^{th}$ edge?
      - Appendix A.1, maybe you mean $\mathcal{S}$ and not $S$ in the definition of $S_B$?
      - In section 5, $\mu'$ is not explained. In other words, you are defining $G$  but not $G'$.
3. I suggest to add the mathematical definition of the error metric in section 6.2
4. Figure 1. seems to be the result of one run of the algorithms only. since the standard deviation is not shown in the plot which is required.
5. The colors of different figures should be consistent.
6. Figure 2 misses definition of one color which makes things confusing.
7. In section 6.2, the paper claims that fair oracle performs similar to submodular oracle. But it is not elaborated or well-motivated why this is important.

**Strengths And Weaknesses:**

# Strengths
1. To consider online setting in for Fair Influence Maximization is novel and  would be interesting for researchers of the field.
2. To map a fairness constraint into a reward function that makes the framework general for many fairness constraints, is an advantage of this framework.
3. Providing theoretical guarantees such as regret analysis gives a good basis for comparison of these approaches with the future algorithms.
4. The problem is well motivated and estimating the edge weights in an online fashion does make sense since there is no appropriate offline data that takes into account fairness constraints (therefore, the existing works lead to poor estimation for minority group edges and they probably would suffer higher regret).
5. The paper is mostly well-written.

# Weaknesses
- There are a number of things that are not clear to me and it would be great if you could elaborate more:
  1. It lacks some explanation about the problem of group-based Fair Influence maximization. i.e. something like: In this setting, we aim to maximize the influence over a number of communities. Meaning that the summation of the influence of all groups should be maximized as opposed to not caring about which part of the graph is gonna receive the resources. This is written in math in section 4, but the intuition is  not really explained in words.
  2. It lacks clarity of some of the steps through the analysis of the work. For instance:
      - It does not explain why equations 12 or 14 hold.
      - Lemma 1, P2 i.e. bounded smoothness for maximin fairness constraint, is not so clear why it holds.
      - The reader has to spend some time to figure out why some of the steps hold while t it is the responsibility of the author. Sometimes it is not so difficult to find out why, and sometimes it is dependent on who the reader is. So, I believe it needs to be explained.
- Regarding the empirical evaluations:
  1.  I think it makes sense that when optimizing for a fair objective, the algorithm which takes fairness into account outperforms one that doesn't. But I am not sure if this is what the paper needs to show. Rather, since the claim is that having an online mechanism to estimate the weights improves the regret when fairness is taken into account, I think it makes more sense to compare the result of some fair offline influence maximization with an online approach.
  2. We are searching for sublinear regret while in the three of figures 1, 2 and 3, some of the approaches are showing linear regret. Although the figures are showing improvement in the initial steps of the algorithm, we don't know which ones are going to reach sublinear regret faster.

---

> ### Author Response · Authors · 2025-04-07
> **Official Reply to Review**
>
> Dear Reviewer q7Vx:
>
> Thank you for your detailed review, and we appreciate your detailed feedback. Bellow, we address the weaknesses pointed out and our manuscript suggestions:
>
> ## Weakness:  clarity of some of the steps
>
> **Concern: Paper clarity, in particular for the proof of equations 12 and 14, and Lemma 1-P2:**
>
> Bellow we include an explanation for equations 12, 14, and Lemma 1, P2:
>
> ### For equation (12)
>
> Given two graph $G$ and $G'$ with the set of edge weights $\mu$ and $\mu'$ respectively, if ~~ $\forall i \in \{1,\ldots,|E|\}, \mu_i \leq \mu_i^{'}$, it is easy to have $\forall C_{i} \in C$:
>
> $$
> {I_{G, C_i}(S)}\leq {I_{G', C_i}(S)}.
> $$
>
> Considering the fact that the graph with a higher weight on each edge will naturally lead to a larger expectation of influenced nodes in the diffusion process.
>
> ### For equation (14)
>
> Suppose, for graphs $G$ and $G'$, the demographic groups that receive the minimum influence proportional to their population are $C_i$ and $C_{i'}$, respectively, such that:
>
> $$
> i = \underset{\forall C_i \in C}{\rm argmin}\frac{I_{G, C_i}(S)}{|C_i|}, \qquad i' = \underset{\forall C_{i'} \in C}{\rm argmin}\frac{I_{G', C_{i'}}(S)}{|C_{i'}|}
> $$
>
> 1. When $i = i'$, we have:
>
> $$
> |C_{i}| = |C_{i'}|, \qquad I_{G, C_{i}}(S) < I_{G', C_{i'}}(S)
> $$
>
> Thus, it is trivial that:
>
> $$
> \frac{I_{G, C_{i}}(S)}{|C_{i}|} = r_\mu(S) \leq r_{\mu'}(S) = \frac{I_{G', C_{i'}}(S)}{|C_{i'}|}
> $$
>
> 2. When $i \neq i'$, , we have:
>
> $$
> \frac{I_{G, C_{i'}}(S)}{|C_{i'}|} \leq \frac{I_{G', C_{i'}}(S)}{|C_{i'}|}
> $$
>
> Since $i'$-th group is the group with minimum proportional influence with respect to its population in $G'$, we have:
>
> $$
> \frac{I_{G, C_i}(S)}{|C_i|} \leq \frac{I_{G, C_{i'}}(S)}{|C_{i'}|}
> $$
> We have revised our paper to be more readable and understandable to every user.
>
> ### Lemma 1, P2
>
> We briefly describe how the bounded smoothness proof process goes. If there is any detail unspecified or causing confusion, we are happy to provide further details.
>
> The proof for bounded smoothness shows that small changes in edge probabilities lead to proportional changes in the influence spread reward. By considering two probability vectors  $\mu$  and  $\mu'$  with a maximum difference of  $\Lambda$  for any edge, the increase in activation from a single edge is at most $ \Lambda$ , which can propagate to at most $|V|$ nodes. For maximin fairness, the objective function increase by 1 at most. Summing over all triggered edges, the total reward change is bounded by  $|\tilde{E}_S|\Lambda$ . Since  $|\tilde{E}_S| \leq |E|$ , the change in reward is proportional to  $\Lambda$, scaled by the graph size  $|E|$  . This ensures stability of the reward function under small probability variations.
>
> We will add more details to make the proof and inference process more clear as suggested.
>
> ---
>
> ## Weakness: Comparison consideration
>
> **Concern: An empirical evaluation to compare regret of offline with online influence maximization**
>
> Additional experimental results in Appendix A.8.5 and A.8.6 demonstrate that even slight perturbations in the activation probabilities fed into offline fair algorithms can result in substantially higher regret. These findings highlight the sensitivity of such algorithms to input accuracy and underscore their inferiority to FOIM in settings with noisy or uncertain activation estimates.
>
> ---
>
> ## Weakness: Sublinear regret and figures 1, 2, and 3
>
> **Concern: Some approaches from figures 1, 2, and 3 show linear regret and it is uncertain which approaches reach sublinear regret faster**
>
> This behavior is expected during the early stages of the online learning process. In the initial iterations, the algorithm is heavily engaged in exploration, which can cause the regret to increase at a nearly linear rate. However, as the algorithm gathers more data and shifts toward exploitation, the growth in regret slows and eventually becomes sublinear, which is consistent with our theoretical guarantees in the CMAB framework.
>
> ---
>
> ## Manuscript Changes:
>
> We have applied the suggested fixes, removed typos, and added the required expressions and legend in Figure 2.
>
> The proof of fair oracle performing similarly to submodular oracles is used to emphasize that employing fair oracle wouldn't introduce severe utility loss.

---

> > ### Comment · Reviewer_q7Vx · 2025-04-15
> >
> > Hi thanks for addressing the concerns.
> >
> > There are still a few points need for discussion:
> > 1. For equation (14): Based on the explanation you gave, the first part of inequality is true for i = i' and the second part for i != i' while all the parts have to hold true for both cases. I have a proof for it in mind, but it is not what explained here. And if we don't have inequality 14, then we cannot get inequality 15. So, this needs to be fixed I think.
> > 2. Lemma1, P2: Thanks, it is definately a good idea to explain on high-level how it works. Still, I don't understand why If C_min != C'_min, the total increase of IM reward function must be smaller than $\frac{I_{G', C_min}(S)}{|C_min|}$ - ... .
> > 3. Comparison consideration: Thanks for the pointer. I think since this is one of the important motivations of FOIM, i.e. existing offline datasets do not assume fairness considerations which leads to inconsistent probability estimations, and therefore, sub-optimal seed selection, we should be able to see that the online approach is really advantages. In the results of A.8.5., first, there is no explanation how one could use offline approaches with online setting. And secondly, I am not sure the reported numbers show meaningful advantage, for instance, what is the standard deviation of the Influenced nodes over multiple runs?
> > 4. Still, Fig 1, looks like to be over only 1 run of experiment. It is important to report the results over multiple runs that are appropriate for your experiment.
> > 5. Sublinear regret and figures 1, 2, and 3: I just realized that the online approaches in these graphs are the lines that are almost horizontal! So, although we cannot see when they converged, they seem to have sublinear regret already. So, that is good.
> > 6. Section 6.2, regarding the added definition, is p_e the same as $\mu$? If so, I suggest using the same notation. Moreover, it is not obvious how to find the absolute discrepancy between two vector values. Do you consider L_1 norm? Do you average? It looks more like averaging because the values are less than 1.
> > 7. I cannot see anywhere if the number of seeds(runs) for each of the plots is reported, and this is very important for reproducibility of the work.

---

> > > ### Author Response · Authors · 2025-04-15
> > > **Official Reply by Authors**
> > >
> > > Thank you for your thoughtful comment and for pointing out the ambiguity in our explanation.
> > >
> > > Here, we provide a formal and coherent proof for Equation (14), integrating both cases into a unified argument. We also include a justification for the upper bound stated in Lemma 2. In addition, we have clarified the experimental setup in Appendix A.8.5, the meaning of the definition in Section 6.2, and the relevant hyperparameter settings.
> > >
> > > We have revised the manuscript accordingly.
> > >
> > > ## Theoretical Side
> > >
> > > >Proof of Equation (14)
> > >
> > > Let $C_i$ be the group minimizing normalized influence in $G$, and $C_{i'}$ the minimizer in $G'$. That is,
> > >
> > > $$
> > > r_\mu(S) = \frac{I_{G, C_i}(S)}{|C_i|}, \quad
> > > r_{\mu'}(S) = \frac{I_{G', C_{i'}}(S)}{|C_{i'}|}.
> > > $$
> > >
> > > We show that $r_\mu(S) \leq r_{\mu'}(S)$ in two steps:
> > >
> > > Step 1. By definition of $C_i$, we have
> > >
> > > $$
> > > \frac{I_{G, C_i}(S)}{|C_i|} \leq \frac{I_{G, C_{i'}}(S)}{|C_{i'}|}.
> > > $$
> > >
> > > Step 2. Since $G'$ induces no less influence than $G$ for any group — that is, the diffusion process in $G'$ activates at least as many nodes as in $G$ under the same seed set — we have
> > >
> > > $$
> > > \frac{I_{G, C_{i'}}(S)}{|C_{i'}|} \leq \frac{I_{G', C_{i'}}(S)}{|C_{i'}|}.
> > > $$
> > >
> > > Combining the two steps, we obtain
> > >
> > > $$
> > > r_\mu(S) = \frac{I_{G, C_i}(S)}{|C_i|} \leq \frac{I_{G', C_{i'}}(S)}{|C_{i'}|} = r_{\mu'}(S).
> > > $$
> > >
> > > >Proof that the change of objective function is bounded by the difference for $C_{\min}$
> > >
> > > Let $C_{\min} = \arg\min_C \frac{I_{G, C}(S)}{|C|}$ be the minimizing group under $G$, and define the objective values as:
> > >
> > > $$
> > > r_{\mu}(S) = \frac{I_{G, C_{\min}}(S)}{|C_{\min}|}, \quad
> > > r_{\mu'}(S) = \min_C \frac{I_{G', C}(S)}{|C|}.
> > > $$
> > >
> > > Even if the minimizing group changes in $G'$, we always have:
> > >
> > > $$
> > > r_{\mu'}(S) \leq \frac{I_{G', C_{\min}}(S)}{|C_{\min}|},
> > > $$
> > >
> > > since $C_{\min}$ is one possible candidate in the minimization.
> > >
> > > Therefore, the change of the objective is upper bounded by:
> > >
> > > $$
> > > r_{\mu'}(S) - r_{\mu}(S)
> > > \leq \frac{I_{G', C_{\min}}(S)}{|C_{\min}|} - \frac{I_{G, C_{\min}}(S)}{|C_{\min}|}.
> > > $$
> > >
> > > ## Experimental Side
> > >
> > > **Response to comparison concern in A.8.5:**
> > >
> > > Thank you for pointing this out. We clarify that the "FW Offline (pert.)" line in Figure 9 represents a fixed seed set selected by the offline Frank-Wolfe method on a perturbed version of the graph, where Gaussian noise $\mathcal{N}(0, 0.05^2)$ is added to each edge's activation probability. This simulates a realistic case where the estimated influence model differs from the true one used during deployment.
> > >
> > > Importantly, this offline seed set is computed once and reused across all rounds. The ground-truth diffusion, however, is simulated on the unperturbed graph, which reflects the true environment. In contrast, online algorithms adapt to the actual observed influence through interaction.
> > >
> > > Since the FW offline seed selection is deterministic under the perturbed graph, its regret curve is a single line and has no standard deviation.
> > >
> > > **Response to $p_e$**
> > >
> > > Thank you for the helpful comments. We clarify as follows:
> > >
> > > - *pₑ* denotes the ground-truth activation probability of edge *e*, which does not depend on the source node or cascade. We use *p̂ₑ,ₜ* for the estimated value at round *t*, and *pₑ* for the true value. The notation is different from $\mu$ as $\mu$ is calculated based on a certain seed set $S$ through cascading structure..
> > >
> > > - The discrepancy is computed as the mean absolute error over observed edges:
> > >
> > >   $$
> > >   \frac{1}{|E|} \sum_{e \in E} \left| \hat{p}_{e,t} - p_e \right|.
> > >   $$
> > >
> > >   This explains why reported values are typically less than 1.
> > >
> > > The number of rounds *t* is dataset-dependent and specified in our hyperparameter table (Section: *Experimental Results – Hyperparameter Settings*): 300 for Twitter, 10 for NBA, and 50 for Amazon.

---

> ### Comment · Reviewer_q7Vx · 2025-04-17
>
> Thanks for taking the feedback into account. There are some of them that are not yet addressed:
>
> 3. Comparison consideration: In the results presented in A.8.5, there are still some clarifications needed:
>     > First, it's unclear how the offline approaches are applied in an online setting.
>
>     > Second, the reported numbers don’t seem to clearly demonstrate a meaningful advantage. For instance, what is the standard deviation of the number of influenced nodes over multiple runs?
>
> $\rightarrow$ If I understand correctly, you are not actually using the offline approaches in an online manner. Instead, you evaluate them once using the initial dataset and then plot the results over time. If that’s the case, I think this should be clarified in the paper.
>
> Also, even if the seed set is fixed, wouldn’t the number of influenced nodes vary across runs due to the probabilistic nature of influence propagation on the graph? If so, that randomness should be reflected in the results.
>
> More specifically regarding Table 3, my concern is that we don’t see any indication of this randomness. Perhaps I’m mistaken, but I would expect different algorithms to show variability in the number of influenced nodes due to this stochasticity. Shouldn't standard deviations be reported to reflect this?
>
> 7. Number of runs/statistical measure: Perhaps I was unclear when referring to the “number of seeds” (since that has a specific meaning in your context). What I meant was that while the plots show variance, there’s no explanation of how many runs were conducted to generate each plot.
>
> For example, are the curves showing the average regret over 10 runs, with shaded regions indicating the standard deviation or confidence intervals? Or is a different statistical measure being used?
>
> It would be helpful to clearly specify what exactly is being plotted and what statistical measure the variance represents in each figure.

---

> > ### Author Response · Authors · 2025-04-17
> > **Official Reply by Authors**
> >
> > Thank you very much for your detailed and thoughtful comments. We appreciate your close reading and would like to clarify the remaining issues regarding Section A.8.5 and Table 3.
> >
> > **Offline Oracle Evaluation**
> >
> > You are correct. Each offline method is run once on a perturbed (noisy) graph to obtain a fixed seed set. This seed set is then evaluated on the true (unperturbed) graph by simulating the influence process under a cascading model. We report the expected number of nodes (so with linear result and without randomness) and no adaptation occurs over time in this setting.
> >
> > **Variability of Experiments**
> >
> > In our experiments, each method is evaluated over 5 or 10 independent runs, and we report both the mean (as the solid line) and the standard deviation (as shaded area) of objective. We have updated Table 3 now to include the mean values along with the corresponding deviation ranges.

---

### Review · Reviewer_4wdB · 2024-12-25

**Summary Of Contributions:**

This paper considers the fair online influence maximization (OIM) problem. The OIM problem has been extensively studied in the literature. This paper first introduces the fairness constraint. It considers different notions of fairness, including the maxmin fairness, diversity fairness, and welfare function. The main contribution of the work is that it verifies the reward function under these notions of fairness satisfies the monontonicity and bounded smoothness properties. Thus applying existing works naturally gets a theoretical guarantee for the regret.

**Audience:**

Yes

**Claims And Evidence:**

Yes

**Requested Changes:**

Please see the above weakness part.

**Strengths And Weaknesses:**

Strength:
1. Considering the fairness in the online influence maximization problem is meaningful and valuable.
2. This paper considers different well-studied fairness definition.
3. This paper shows that the reward function under these fairness constraint satisfies monontonicity and bounded smoothness.
4. The work compares the empirical performances of classic OIM algorithms under different fairness oracles.

Weakness:
One of the key concerns is the correctness of the results. In Section 5.2, the regret depends on $1/\mu^*$ (or $1/\mu_i$) where $\mu^*$ is the minimum weight of the edge. But in Chen et al. (2014), this term is $1/p^*$ (or $1/p_i$) which represents the minimum triggering probability of edges that can be exponentially smaller than $\mu^*$. Please check this.

The definition of $\Delta_{\min}, \Delta_{\max}...$ are all not introduced before used.

The writing is not clear enough and can still be improved. Some important definitions are missing.
1. In Section 4, the definition of group (community) is not introduced. Should different groups be disjoint or not?
2. If different groups are disjoint, (4) is equivalent to (1).
3. In (11), the group is denoted as $c$ while it is denoted as $C_i$ in previous parts. It is not clear what is the definition of $u$. The definition should be consistent across the full paper.
4. When introducing the fairness constraint, what is the formal definition of the regret?
5. The end of Lemma 2: missing a period
6. More introduction of the oracles should be provided. Such as the value of $\alpha$ and $\beta$ for the oracles introduced.

Discussion:
For the OIM problem, the state-of-the-art result depends on the TPM condition instead of the bounded smoothness condition [Wang & Chen, 2017; Wen et al., 2016)]. Do the authors try to validate whether the reward function satisfy the TPM condition?

---

> ### Author Response · Authors · 2025-04-07
> **Official Reply to Review**
>
> Dear Reviewer 4wdB:
>
> Thank you for your detailed review and for highlighting the key strengths of our paper. Bellow, you can find our responses to the questions, concerns, and suggestions made.
>
> ## Weakness: Correctness of regret result from section 5.2
>
> **Concern: Regret depends on $1/\mu^{*}$ despite result from Chen et al. (2014)**
>
> In our previous discussion, we inadvertently confused the concepts of $\mu$ and $p$.
> According to Chen et al. (2014), $\mu$ represents the ground-truth activation probability of an edge, while $p$ denotes the triggering probability. However, Theorem 1 specifically refers to $p$, not $\mu$. Despite this misuse of notation in our derivation, the final results remain unaffected because the subsequent regret analysis can be equivalently adjusted. We have corrected this issue in our improved manuscript version.
>
> ---
>
> ## Weakness: Definition of $\Delta_{max}$ and $\Delta_{min}$:
>
> **Concern: Definition of $\Delta_{max}$ and $\Delta_{min}$ not introduced before using**
>
> In maintaining a clear format and notation reference, we define $\Delta_{max}$ and $\Delta_{min}$ in Appendix A.1 and we have also revised the main content to briefly introduce these two concepts.
>
> ---
>
> ## Weakness: Clarification on Group (Community) Definition and Disjointedness
>
> **Concern: (1) Definition of group (community) is not introduced. Should groups be disjoint? (2) If different groups are disjoint, (4) is equivalent to (1).**
>
> Whether the groups are disjoint is determined by the specific fairness constraint being imposed. In many fairness formulations--such as maximin, diversity, or welfare-based criteria--the groups are not required to be disjoint.For example, in maximin fairness (e.g., Tsang et al., 2019), overlapping groups are explicitly allowed, as the fairness objective only depends on the group with the minimal utility. Such overlapping does not affect the feasibility guarantees (e.g., bounded smoothness or monotonicity).
>
> ---
>
> ## Weakness: Definitions, Typos, and Introduction to Oracles:
>
> **Concern: (3) Inconsistency of notation; (4) Definition of regret after fairness constraint introduction; (5) lemma 2 typo; and (6) Introduction to oracles**
>
> We have fixed the typos and defined regret in Eq (3). We gave an example of $(\alpha, \beta)$ approximation in Definition 1 in revised version and clarified them more clearly. In short, {$(\alpha, \beta)$-approximation} refers to an oracle that, with probability $\beta$, returns a solution whose expected reward is at least an $\alpha$ fraction of the optimal.
>
> ---
>
> ## Discussion: reward function and TPM condition
>
> Thank you for your suggestion and discussion on the TPM condition and our fair objective function. Bellow, you can find our proposed analysis, using *Maximin Fairness* as an example. We find that TPM can be satisfied by some fair objective function. However, not all fair metrics seem to be capable of being adjusted. Below, we prove the infeasibility of Diversity Fairness and feasibility proof of Maximin Fairnes, respectively. Same ontent has also been updated in our revised paper.
>
> > Proof: Diversity Fairness
>
> A seed set $ S $ barely satisfies the diversity constraint under $ \mu $, yielding positive reward.
> A slight change $ \epsilon $ to one edge (new $ \mu' $) may break the constraint, dropping the reward to zero.
>
> Reward change: constant
> TPM bound:
> $$
> B \sum_i p^S_i |\mu_i - \mu'_i| \to 0 \text{ as } \epsilon \to 0
> $$
>
> Thus, the TPM inequality fails for any constant $ B $, so the diversity objective violates the TPM condition.
>
> > Proof: Maximin Fairness
>
> Let
> $$
> r^{\text{maxmin}}_{\mu}(S) = \frac{IG(C^*, \mu)}{|C^*|}
> $$
> for the group $C^*$ minimizing the objective.
>
> Then for any group $ C$:
> $$
> \left| \frac{IG_C(S, \mu)}{|C|} - \frac{IG_C(S, \mu')}{|C|} \right| = \frac{1}{|C|} \cdot |IG_C(S, \mu) - IG_C(S, \mu')|
> $$
>
> Assume under IC model:
> $$
> |IG_C(S, \mu) - IG_C(S, \mu')| \le \sum_{i \in E_S} p^S_i \cdot |\mu_i - \mu'_i|
> $$
>
> So,
> $$
> \left| \frac{IG_C(S, \mu)}{|C|} - \frac{IG_C(S, \mu')}{|C|} \right| \le \frac{1}{|C|} \sum_{i \in E_S} p^S_i \cdot |\mu_i - \mu'_i|
> $$
>
> Take max over $C \in \mathcal{C} $:
> $$
> |r_{\mu}(S) - r_{\mu'}(S)| = \max_C \frac{1}{|C|} \sum_{i \in E_S} p^S_i |\mu_i - \mu'_i|
> $$
>
>
> Let
> $$
> B = \max_{C} \frac{1}{|C|}
> $$
>
> Then
> $$
> \left| r_{\mu}(S) - r_{\mu'}(S) \right| \le B \cdot \sum_{i \in E_S} p^S_i \cdot |\mu_i - \mu'_i|
> $$
>
> This proves the TPM condition.
>
>
> ---
>
> We hope our detailed responses address all of your concerns. your feedback has been invaluable in improving the clarity, depth, and scope of our work.

---

### Review · Reviewer_3xCC · 2025-03-27

**Summary Of Contributions:**

The paper proposes FOIM (Fair Online Influence Maximization) to select the most influential seeds in social networks under the fairness constraints. Different with traditional methods, FOIM learns edge activation probabilities dynamically in an online setting and ensures equitable influence across demographic groups under certain fairness metrics. It integrates a combinatorial multi-armed bandit (CMAB) approach with fairness-aware selection criteria. The authors adopts traditional fairness metrics, such as maximin and diversity fairness, providing theoretical guarantees with sublinear regret bounds. Experiments conducted on real datasets demonstrate the performance of the proposed FOIM.

**Audience:**

Yes

**Claims And Evidence:**

Yes

**Requested Changes:**

please see weaknesses above

**Strengths And Weaknesses:**

Strengths:
* The paper is clearly written and easy to follow.
* It provides theoretical analysis on fairness regret and the properties of fairness metrics.
* Several experiments are conducted using real-world datasets.
* The fair online influence maximization problem is practically relevant.

Weaknesses:

* Key components of the proposed FOIM algorithm (e.g., CMAB) are borrowed from previous work. The theoretical results, specifically the (α, β)-approximation guarantee, also heavily rely on existing literature.
* The time complexity of the proposed algorithm is not provided. Is the complexity exponential or polynomial? Please clarify the exact form.
* For FOIM using the welfare function, since the expression depends on α, it would be clearer to denote it explicitly as $r_\mu(S, \alpha)$
* It would be helpful to report the running times for the experiments across different iterations.

Questions:

* In Equation (8), why does $I_{G, [C_i]}$ depend only on the value k and not on the specific nodes selected?
* In Figure 1, why is the regret of IMFB always zero in the left plot?

---

> ### Author Response · Authors · 2025-04-07
> **Official Reply to Review**
>
> Dear Reviewer 3xCC,
>
> Thank you for your detailed and constructive review. We appreciate your time and effort in evaluating our work. Below, we provide a point-by-point response to your comments and questions.
>
> ## Weakness: Relationship with exisiting works
>
> **Concern: Reliance on existing literature for constructing FOIM framework**
>
> While it is true that FOIM leverages the CMAB framework, we chose this framework deliberately because it has been widely recognized in the literature as a simple yet effective solution paradigm for Online Influence Maximization (OIM) problems. In fact, to the best of our knowledge, most existing OIM works are built upon the CMAB framework due to its flexibility. Our adoption of CMAB ensures that FOIM is not only grounded in a solid foundation but also scalable to all prior works, enabling a general recipe to upgrade existing OIM algorithms into their fair counterparts.
>
> More importantly, incorporating fairness constraints introduces non-trivial theoretical challenges. Our $(\alpha, \beta)$-approximation guarantee is derived under fairness-aware objectives (e.g., maximin and diversity fairness) with proof needed for monotonicity and bounded smoothness, which require new analysis beyond standard CMAB results. Additionally, our extensive experiments on real-world datasets validate FOIM's superior performance under fairness constraints. We will clarify these contributions more explicitly in the revised version.
>
> ---
>
> ## Weakness: Time Complexity and Running Time of Algorithm
>
> **Concern: (1) What is the time complexity of the proposed algorithm? (4) Can you report the running times for the experiments accross different iterations?**
>
> Thank you for pointing this out. We have added a detailed time complexity analysis as well as recorded experiment running times in Appendix A.12. We apologize for not providing them in main paper due to page limit.(1) In brief, FOIM operates within the CMAB framework, and its time complexity largely depends on the complexity of the underlying Oracle used for seed selection. For instance, using the CUCB algorithm, the overall complexity per round is $O(T(|E|\log|E| + Oracle_\text{time}))$ . For fair oracles like the Frank-Wolfe-based one, the complexity is polynomial and given by $O\left(\frac{(|E| + m)^2 k^2 b \log^2 |V|}{\epsilon \delta}\right)$. (2) We have also recorded the running time in Appendix A.12, as in Table 5.
>
> ---
>
> ## Question on Equation 8:
>
> **Concern: Why does$I_{G[C_i]}(k_i)$ depend only on the value $k$ and not on the specific nodes selected?**
>
> Thank you for the insightful question. In Equation (8), the term $I_{G[C_i]}(k_i)$ refers to the expected influence achieved by selecting $k_i$ seed nodes uniformly at random within the sub-graph $G[C_i]$. Since this is an expectation over all possible sets of $k_i$ nodes, it depends only on the value of $k_i$, not on any specific node selection.
>
> ---
>
> ## Question on Figure 1:
>
> **Concern: Why is the regret of IMFB always zero in the left plot?**
>
> Thank you for the question. In the left plot of Figure 1 (MIP Maximin Regret), the regret of IMFB appears to be zero mainly because its regret values are significantly lower--by more than an order of magnitude--compared to other methods. This creates a visual impression that its regret is flat or near-zero, though in reality, it is _non-zero and increasing slightly_ over iterations.
> This contrast further highlights the effectiveness and reliability of FOIM, which maintains competitive or better regret.
>
> ---
>
> We hope our detailed responses address all of your concerns. your feedback has been invaluable in improving the clarity, depth, and scope of our work.

---

### Author Response · Authors · 2025-02-17
**Follow-up on third review**

This is a follow-up regarding our manuscript *Fair Online Influence Maximization* submitted to TMLR. Currently, two reviews have been completed, but as TMLR requires three for a comprehensive evaluation, we would like to kindly check on the status of the third review.

We are ready to draft a detailed response to the reviewers’ comments; however, we would like to ensure a thorough and constructive discussion, which depends on the completion of the third review. We truly appreciate the time and effort of the editorial team and reviewers in facilitating the process and look forward to the next steps.

---

### Decision · Action_Editor_eLrK · 2025-05-20

**Recommendation:** Accept with minor revision

**Comment:**

Two reviewers have given the recommendation "Leaning Accept", and one reviewer has given the recommendation "Accept". After reading the paper, the reviews, and the discussion, I recommend to accept this paper with minor revision.

Specifically, I recommend the authors to make the following revisions:

1) Please make all the revisions promised in the author responses below.

2) Please improve the writing of Section 6. In particular, please add a paragraph at the beginning of Section 6 to summarize the experiment results. Also, please make the figures larger so they are easier to read.

3) In Algorithm 1, the authors refer to $\mathcal{A}$ as "CMAB algorithm". I do not think this is an appropriate name. Specifically, $\mathcal{A}$ only estimates the edge weights, but does not choose the seed nodes, so it is just part of a "CMAB algorithm". I think probably we should call it "edge weight estimator", and clarify that it can be an optimistic edge weight estimator.

4) There seems to be a typo in the Reference. Two papers with the same title "Online Influence Maximization under Independent Cascade Model ..." but different authors have been cited.

5) possible typo: under equation 1, "and $I_G(S)$ is the maximum number of influence nodes ...", I think "maximum" should be "expected".

**Audience:**

Yes, I believe that some audience of TMLR will be interested in findings of this paper.

**Claims And Evidence:**

Yes. To the best of my knowledge, the claims made in the submission are supported by accurate, convincing, and clear evidence.